# MaskGCT: Zero-Shot Text-to-Speech with Masked Generative Codec Transformer

**Yuancheng Wang**[1], **Haoyue Zhan**[2], **Liwei Liu**[1], **Ruihong Zeng**[2], **Haotian Guo**[1],
**Jiachen Zheng**[1], **Qiang Zhang**[2], **Shunsi Zhang**[2], **Xueyao Zhang**[1], **Zhizheng Wu**[1]
[1]The Chinese University of Hong Kong, Shenzhen
[2]Guangzhou Quwan Network Technology
`yuanchengwang@link.cuhk.edu.cn, zhizhengwu@cuhk.edu.cn`

## ABSTRACT

The recent large-scale text-to-speech (TTS) systems are usually grouped as autoregressive and non-autoregressive systems. The autoregressive systems implicitly model duration but exhibit certain deficiencies in robustness and lack of duration controllability. Non-autoregressive systems require explicit alignment information between text and speech during training and predict durations for linguistic units (e.g. phone), which may compromise their naturalness. In this paper, we introduce **Mask**ed **G**enerative **C**odec **T**ransformer (MaskGCT), *a fully non-autoregressive TTS model that eliminates the need for explicit alignment information between text and speech supervision, as well as phone-level duration prediction*. MaskGCT is a two-stage model: in the first stage, the model uses text to predict semantic tokens extracted from a speech self-supervised learning (SSL) model, and in the second stage, the model predicts acoustic tokens conditioned on these semantic tokens. MaskGCT follows the *mask-and-predict* learning paradigm. During training, MaskGCT learns to predict masked semantic or acoustic tokens based on given conditions and prompts. During inference, the model generates tokens of a specified length in a parallel manner. Experiments with 100K hours of in-the-wild speech demonstrate that MaskGCT outperforms the current state-of-the-art zero-shot TTS systems in terms of quality, similarity, and intelligibility. Audio samples are available at `https://maskgct.github.io/`. We release our code and model checkpoints at `https://github.com/open-mmlab/Amphion/blob/main/models/tts/maskgct`.

## 1 INTRODUCTION

In recent years, large-scale zero-shot text-to-speech (TTS) systems [1, 2, 3, 4, 5, 6, 7, 8, 9, 10] have achieved significant improvements by scaling data and model sizes, including both autoregressive (AR) [1, 2, 3, 4, 5, 6] and non-autoregressive (NAR) models [7, 8, 9, 10]. However, both AR-based and NAR-based systems still exhibit some shortcomings. In particular, AR-based TTS systems typically quantize speech into discrete tokens and then use decoder-only models to autoregressively generate these tokens, which offer diverse prosody but also suffer from problems such as poor robustness and slow inference speed. NAR-based models, typically based on diffusion [7, 8], flow matching [9], or GAN [10], require explicit text and speech alignment information as well as the prediction of phone-level duration, resulting in a complex pipeline and producing more standardized but less diverse speech.

Recently, masked generative transformers, a class of generative models, have achieved significant results in the fields of image [11, 12, 13], video [14, 15], and audio [16, 17, 18] generation, demonstrating potential comparable to or superior to autoregressive models or diffusion models. These models employ a mask-and-predict training paradigm and utilize iterative parallel decoding during inference. Some previous works have attempted to introduce masked generative models into the field of TTS. SoundStorm [19] was the first attempt to use a masked generative transformer to predict multi-layer acoustic tokens extracted from SoundStream, conditioned on speech semantic tokens; however, it needs to receive the semantic tokens of an AR model as input. Thus, SoundStorm is more of an acoustic model that converts semantic tokens into acoustic tokens and does not fully utilize the

powerful generative potential of masked generative models. NaturalSpeech 3 [8] decomposes speech into discrete token sequences representing different attributes through special designs and generates tokens representing different attributes through masked generative models. However, it still needs speech-text alignment supervision and phone-level duration prediction.

In this work, we propose MaskGCT, ***a fully non-autoregressive model for text-to-speech synthesis that uses masked generative transformers without requiring text-speech alignment supervision and phone-level duration prediction***. MaskGCT is a two-stage system, both stages are trained using the *mask-and-predict learning* paradigm. The first stage, the text-to-semantic (T2S) model, predicts masked semantic tokens with in-context learning, using text token sequences and prompt speech semantic token sequences as the prefix, without explicit duration prediction. The second stage, the semantic-to-acoustic (S2A) model, utilizes semantic tokens to predict masked acoustic tokens extracted from an RVQ-based speech codec with prompt acoustic tokens. During inference, MaskGCT can generate semantic tokens of various specified lengths with a few iteration steps given a sequence of text. In addition, we train a VQ-VAE [20] to quantize speech self-supervised learning embedding, rather than using k-means to extract semantic tokens that is common in previous work. This approach minimizes the information loss of semantic features even with a single codebook. We also explore the scalability of our methods beyond the zero-shot TTS task, such as speech translation (cross-lingual dubbing), speech content editing, voice conversion, and emotion control, demonstrating the potential of MaskGCT as a foundational model for speech generation. Appendix A.6 shows a comparison between MaskGCT and some previous works.

Our experiments demonstrate that MaskGCT has achieved performance comparable to or superior to that of existing models in terms of speech quality, similarity, prosody, and intelligibility. Specifically, (1) It achieves comparable or better quality and naturalness than the ground truth speech across three benchmarks (LibriSpeech, SeedTTS *test-en*, and SeedTTS *test-zh*) in terms of CMOS. (2) It achieves human-level similarity between the generated speech and the prompt speech, with improvements of +0.017, -0.002, and +0.027 in SIM-O and +0.28, +0.32 and +0.25 in SMOS for LibriSpeech, SeedTTS *test-en*, and SeedTTS *test-zh*, respectively. (3) It achieves comparable intelligibility in terms of WER across the three benchmarks and demonstrates stability within a reasonable range of speech duration, which also indicates the diversity and controllability of the generated speech.

In summary, we propose a non-autoregressive zero-shot TTS system based on masked generative transformers and introduce a speech discrete semantic representation by training a VQ-VAE on speech self-supervised representations. Our system achieves human-level similarity, naturalness, and intelligibility by scaling data to 100K hours of in-the-wild speech, while also demonstrating high flexibility, diversity, and controllability. We investigate the scalability of our system across various tasks, including cross-lingual dubbing, voice conversion, emotion control, and speech content editing, utilizing zero-shot learning or post-training methods. This showcases the potential of our system as a foundational model for speech generation.

## 2 RELATED WORK

**Large-scale TTS.** Traditional TTS systems [21, 22, 23, 24, 25] are trained to generate speech from a single speaker or multiple speakers using hours of high-quality transcribed training data. Modern large-scale TTS systems [1, 2, 3, 4, 5, 6] aim to achieve zero-shot TTS (synthesizing speech for unseen speakers with speech prompts) by scaling both the model and data size. These systems can be mainly divided into AR-based and NAR-based categories. For AR-based systems: SpearTTS [1] utilizes three AR models to predict semantic tokens from text, coarse-grained acoustic tokens from semantic tokens, and fine-grained acoustic tokens from coarse-grained tokens. VALL-E [2] predicts the first layer of acoustic tokens extracted from EnCodec [26] using an AR codec language model, and the final layers with a NAR model. VoiceCraft [5] employs a single AR model to predict multi-layer acoustic tokens in a delayed pattern [27]. BASETTS [3] predicts novel speech codes extracted from WavLM features and uses a GAN model for waveform reconstruction. For NAR-based systems: NaturalSpeech 2 [7] employs latent diffusion to predict the latent representations from a codec model [28]. VoiceBox [9] and P-Flow [29] use flow matching and in-context learning to predict mel-spectrograms. MegaTTS [10] utilizes a GAN to predict mel-spectrograms, while an AR model predicts phone-level prosody codes. NaturalSpeech 3 [8] employs a unified framework based on discrete diffusion models to predict discrete representations of different speech attributes. However,

these NAR systems need to predict phoneme-level duration, leading to a complex pipeline and more standardized generative results. SimpleSpeech [30], DiTTo-TTS [31], and E2 TTS [32] are also NAR-based models that do not require precise alignment information between text and speech, nor do they predict phoneme-level duration. We discuss these concurrent works in Appendix K.

**Masked Generative Model.** Masked generative transformers, a class of generative models, achieve significant results and demonstrate potential comparable to or superior to that of autoregressive models or diffusion models in the fields of image [11, 12, 13, 33], video [14, 15], and audio [16, 17, 18, 19] generation. MaskGIT [11] is the first work to use masked generative models for both unconditional and conditional image generation. Subsequently, Muse [12] leverages rich text to achieve high-quality and diverse text-to-image generation within the same framework. MAGVIT-v2 [15] employs masked generative models with novel lookup-free quantization, outperforming diffusion models in image and video generation. Recently, some efforts have been made to adapt masked generative models to the field of audio. SoundStorm [19] takes in the semantic tokens from AudioLM and utilizes this generative paradigm to generate tokens for a neural audio codec [28]. VampNet [16] and MAGNeT [18] apply masked generative models for music and audio generation, while MaskSR [17] extends these models for speech restoration.

**Discrete Speech Representation.** Speech representation is a crucial aspect of speech generation. Early works [22, 24] typically utilized mel-spectrograms as the modeling target. Recently, some large-scale TTS systems [2, 8] have shifted to using discrete speech representations. Discrete speech representation can be primarily divided into two types: semantic discrete representation and acoustic discrete representation[1]. Semantic discrete representations are mainly extracted from various speech SSL models [34, 35, 36] using quantization methods such as k-means. Acoustic discrete representations, on the other hand, are usually obtained by training a VQ-GAN model [20] with the goal of waveform reconstruction, as seen in speech codecs [26, 28, 37]. Semantic discrete representation typically shows a stronger correlation with text, whereas acoustic discrete representation more effectively reconstructs audio. Consequently, some two-stage TTS models predict both semantic and acoustic tokens. FACodec [8] is a novel speech codec that disentangles speech into subspaces of different attributes, including content, prosody, timbre, and acoustic details.

# 3 METHOD

## 3.1 BACKGROUND: NON-AUTOREGRESSIVE MASKED GENERATIVE TRANSFORMER

Given a discrete representation sequence $\mathbf{X}$ of some data, we define $\mathbf{X}_t = \mathbf{X} \odot \mathbf{M}_t$ as the process of masking a subset of tokens in $\mathbf{X}$ with the corresponding binary mask $\mathbf{M}_t = [m_{t,i}]_{i=1}^N$. Specifically, this involves replacing $x_i$ with a special [MASK] token if $m_{t,i} = 1$, and otherwise leaving $x_i$ unmasked if $m_{t,i} = 0$. Here, each $m_{t,i}$ is independently and identically distributed according to a Bernoulli distribution with parameter $\gamma(t)$, where $\gamma(t) \in (0, 1]$ represents a mask schedule function (for example, $\gamma(t) = \sin(\frac{\pi t}{2T}), t \in (0, T]$). We denote $\mathbf{X}_0 = \mathbf{X}$. The non-autoregressive masked generative transformers are trained to predict the masked tokens based on the unmasked tokens and a condition $\mathbf{C}$. This prediction is modeled as $p_\theta(\mathbf{X}_0|\mathbf{X}_t, \mathbf{C})$. The parameters $\theta$ are optimized to minimize the negative log-likelihood of the masked tokens:

$$\mathcal{L}_{\text{mask}} = \mathbb{E}_{\mathbf{X} \in \mathcal{D}, t \in [0,T]} - \sum_{i=1}^N m_{t,i} \cdot \log(p_\theta(x_i|\mathbf{X}_t, \mathbf{C})).$$

At the inference stage, we decode the tokens in parallel through iterative decoding. We start with a fully masked sequence $\mathbf{X}_T$. Assuming the total number of decoding steps is $S$, for each step $i$ from 1 to $S$, we first sample $\hat{\mathbf{X}}_0$ from $p_\theta(\mathbf{X}_0|\mathbf{X}_{T-(i-1)\cdot\frac{T}{S}}, \mathbf{C})$. Then, we sample $\lfloor N \cdot \gamma(T - i \cdot \frac{T}{S}) \rfloor$ tokens based on the confidence score to remask, resulting in $\mathbf{X}_{T-i\cdot\frac{T}{S}}$, where $N$ is the total number of tokens in $\mathbf{X}$. The confidence score for $\hat{x}_i$ in $\hat{\mathbf{X}}_0$ is assigned to $p_\theta(x_i|\mathbf{X}_{T-(i-1)\cdot\frac{T}{S}}, \mathbf{C})$ if $x_{T-(i-1)\cdot\frac{T}{S},i}$ is a [MASK] token; otherwise, we set the confidence score of $\hat{x}_i$ to 1, indicating that tokens already unmasked in $\mathbf{X}_{T-(i-1)\cdot\frac{T}{S}}$ will not be remasked. Particularly, we choose $\lfloor N \cdot \gamma(T - i \cdot \frac{T}{S}) \rfloor$ tokens with the lowest confidence scores to be masked.

---

[1]We give a more detailed discussion about the definitions of "semantic" and "acoustic" in Appendix B.

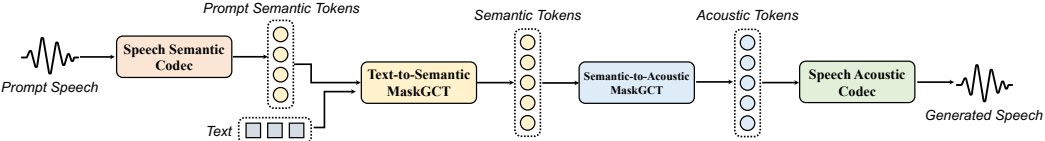

Figure 1: An overview of the proposed two-stage MaskGCT framework. It consists of four main components: (1) a speech semantic representation codec converts speech to semantic tokens; (2) a text-to-semantic model predicts semantic tokens with text and prompt semantic tokens; (3) a semantic-to-acoustic model predicts acoustic tokens conditioned on semantic tokens; (4) a speech acoustic codec reconstructs waveform from acoustic tokens.

The masked generative modeling paradigm was first introduced in [11], and subsequent work such as [33] has further explored it under the perspective of discrete diffusion.

## 3.2 MODEL OVERVIEW

An overview of the MaskGCT framework is presented in Figure 1. Following [2, 19, 38], MaskGCT is a two-stage TTS system. The first stage uses text to predict speech semantic representation tokens, which contain most information of content and partial information of prosody. The second stage model is trained to learn more acoustic information. Unlike previous works [1, 2, 19, 38] use an autoregressive model for the first stage, MaskGCT utilizes the non-autoregressive masked generative modeling paradigm for both the two stages without text-speech alignment supervision and phone-level duration prediction: (1) For the first stage model, we trained a model to learn $p_{\theta_{s1}}(\mathbf{S}|\mathbf{S}_t, (\mathbf{S}^p, \mathbf{P}))$, where $\mathbf{S}$ is the speech semantic representation token sequence obtained from a speech semantic representation codec (we introduce in 3.2.1), $\mathbf{S}^p$ is the prompt semantic token sequence, and $\mathbf{P}$ is the text token sequence. $\mathbf{S}^p$ and $\mathbf{P}$ are the condition for the first stage model. (2) The second stage model is trained to learn $p_{\theta_{s2}}(\mathbf{A}|\mathbf{A}_t, (\mathbf{A}^p, \mathbf{S}))$, where $\mathbf{A}$ is the multi-layer acoustic token sequence from a speech acoustic codec like [26, 28]. Our second stage model is similar to SoundStorm [19]. We give more details about the four parts in the following sections.

### 3.2.1 SPEECH SEMANTIC REPRESENTATION CODEC

Discrete speech representations can be divided into semantic tokens and acoustic tokens. Generally, semantic tokens are obtained by discretizing features from speech self-supervised learning (SSL). Previous two-stage, large-scale TTS systems [1, 19, 38] typically first use text to predict semantic tokens, and then employ another model to predict acoustic tokens or features. This is because semantic tokens have a stronger correlation with text or phonemes, which makes predicting them more straightforward than directly predicting acoustic tokens. Commonly, previous works have used k-means to discretize semantic features to obtain semantic tokens; however, this method can lead to a loss of information. This loss may complicate the accurate reconstruction of high-quality speech or the precise prediction of acoustic tokens, especially for tonally rich languages. For example, our early experiments demonstrate the challenges of accurately predicting acoustic tokens to achieve proper prosody for Chinese using semantic tokens obtained via k-means. We give more experimental explorations in Section 4.4. Therefore, we need to discretize semantic representation features while minimizing information loss. Inspired by [39], we train a VQ-VAE model to learn a vector quantization codebook that reconstructs speech semantic representations from a speech SSL model. For a speech semantic representation sequence $\mathbf{S} \in \mathbb{R}^{T \times d}$, the vector quantizer quantizes the output of the encoder $\mathcal{E}(\mathbf{S})$ to $\mathbf{E}$, and the decoder reconstructs $\mathbf{E}$ back to $\hat{\mathbf{S}}$. We optimize the encoder and the decoder using a reconstruction loss between $\mathbf{S}$ and $\hat{\mathbf{S}}$, employ codebook loss to optimize the codebook and use commitment loss to optimize the encoder with the straight-through method [20]. The total loss for training the semantic representation codec can be written as:

$$\mathcal{L}_{\text{total}} = \frac{1}{Td}(\lambda_{\text{rec}} \cdot ||\mathbf{S} - \hat{\mathbf{S}}||_1 + \lambda_{\text{codebook}} \cdot ||\text{sg}(\mathcal{E}(\mathbf{S})) - \mathbf{E}||_2 + \lambda_{\text{commit}} \cdot ||\text{sg}(\mathbf{E}) - \mathcal{E}(\mathbf{S})||_2).$$

where sg means stop-gradient.

In detail, we utilize the hidden states from the 17th layer of W2v-BERT 2.0 [34] as the semantic features for our speech encoder. The encoder and decoder are composed of multiple ConvNext [40]

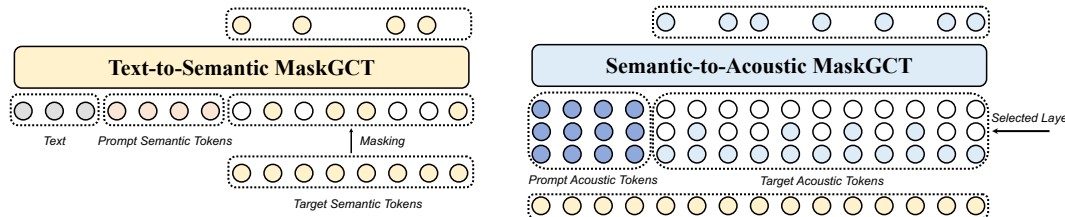

Figure 2: An overview of training diagram of the T2S (left) and S2A (right) models. The T2S model is trained to predict masked semantic tokens with text and prompt semantic tokens as the prefix. The S2A model is trained to predict masked acoustic tokens of a random layer conditioned on prompt acoustic tokens, semantic tokens, and acoustic tokens of the previous layers.

blocks. Following the methods of improved VQ-GAN [41] and DAC [37], we use factorized codes to project the output of the encoder into a low-dimensional latent variable space. The codebook contains 8,192 entries, each of dimension 8. Further details about the model architecture are provided in Appendix A.4.

### 3.2.2 TEXT-TO-SEMANTIC MODEL

Based on the previous discussion, we employ a non-autoregressive masked generative transformer to train a text-to-semantic (T2S) model, instead of using an autoregressive model or any text-to-speech alignment information. During training, we randomly extract a portion of the prefix of the semantic token sequence as the prompt, denoted as $\mathbf{S}^p$. We then concatenate the text token sequence $\mathbf{P}$ with $\mathbf{S}^p$ to form the condition. We simply add $(\mathbf{P}, \mathbf{S}^p)$ as the prefix sequence to the input masked semantic token sequence $\mathbf{S}_t$ to leverage the in-context learning ability of language models. We use a Llama-style [42] transformer as the backbone of our model, incorporating gated linear units with GELU [43] activation, rotation position encoding [44], etc., but replacing causal attention with bidirectional attention. We also use adaptive RMSNorm [45], which accepts the time step $t$ as the condition.

During inference, we generate the target semantic token sequence of any specified length conditioned on the text and the prompt semantic token sequence. In this paper, we also train a flow matching [46] based duration prediction model to predict the total duration conditioned on the text and prompt speech duration, leveraging in-context learning. More details can be found in Appendix A.5.

### 3.2.3 SEMANTIC-TO-ACOUSTIC MODEL

We also train a semantic-to-acoustic (S2A) model using a masked generative codec transformer conditioned on the semantic tokens. Our semantic-to-acoustic model is based on SoundStorm [19], which generates multi-layer acoustic token sequences. Given $N$ layers of the acoustic token sequence $\mathbf{A}^{1:N}$, during training, we select one layer $j$ from 1 to $N$. We denote the $j$th layer of the acoustic token sequence as $A^j$. Following the previous discussion, we mask $A^j$ at the timestep $t$ to get $\mathbf{A}_t^j$. The model is then trained to predict $\mathbf{A}^j$ conditioned on the prompt $\mathbf{A}^p$, the corresponding semantic token sequence $\mathbf{S}$, and all the layers smaller than $j$ of the acoustic tokens. This can be formulated as $p_{\theta_{s2a}}(\mathbf{A}^j | \mathbf{A}_t^j, (\mathbf{A}^p, \mathbf{S}, \mathbf{A}^{1:j-1}))$. We sample $j$ according to a linear schedule $p(j) = 1 - \frac{2j}{N(N+1)}$. For the input of the S2A model, since the number of frames in the semantic token sequence is equal to the sum of the frames in the prompt acoustic sequence and the target acoustic sequence, we simply sum the embeddings of the semantic tokens and the embeddings of the acoustic tokens from layer 1 to $j$. During inference, we generate tokens for each layer from coarse to fine, using iterative parallel decoding within each layer. Figure 2 shows a simplified training diagram of the T2S and S2A models.

### 3.2.4 SPEECH ACOUSTIC CODEC

Speech acoustic codec is trained to quantize speech waveform to multi-layer discrete tokens while aiming to preserve all the information of the speech as soon as possible. We follow the residual vector quantization (RVQ) method to compress the 24K sampling rate speech waveform into discrete tokens

of 12 layers. The codebook size of each layer is 1,024 and the codebook dimension is 8. The model architectures, discriminators, and training losses follow DAC [37], except that we use the Vocos [47] architecture as the decoder for more efficient training and inference. Figure 5 shows the comparison between the semantic codec and acoustic codec.

## 3.3 OTHER APPLICATIONS

MaskGCT can accomplish tasks beyond zero-shot TTS, such as duration-controllable speech translation (cross-lingual dubbing), emotion control, speech content editing, and voice conversion with simple modifications or the assistance of external tools, demonstrating the potential of MaskGCT as a foundational model for speech generation. We provide more details in Appendix F, G, H, I.

## 4 EXPERIMENTS AND RESULTS

### 4.1 EXPERIMENTAL SETTINGS

**Datasets.** We use the Emilia [48] dataset to train our models. Emilia is a multilingual and diverse in-the-wild speech dataset designed for large-scale speech generation. In this work, we use English and Chinese data from Emilia, each with 50K hours of speech (totaling 100K hours). We evaluate our zero-shot TTS models with three benchmarks: (1) LibriSpeech [49] *test-clean*, a widely used test set for English zero-shot TTS. (2) SeedTTS *test-en*, a test set introduced in Seed-TTS [6] of samples extracted from English public corpora, includes 1,000 samples from the Common Voice dataset [50]. (3) SeedTTS *test-zh*, a test set introduced in Seed-TTS of samples extracted from Chinese public corpora, includes 2,000 samples from the DiDiSpeech dataset [51]. We also scale the training dataset to six languages to support multilingual zero-shot TTS. We provide additional experimental details and evaluation results about multilingual zero-shot TTS in Appendix E.

**Evaluation Metrics.** We use both objective and subjective metrics to evaluate our models. For the objective metrics, we evaluate speaker similarity (SIM-O), robustness (WER), and speech quality (FSD). Specifically, for speaker similarity, we compute the cosine similarity between the WavLM TDNN[2] [36] speaker embedding of generated samples and the prompt. For Word Error Rate (WER), we use a HuBERT-based[3] ASR model for LibriSpeech *test-clean*, Whisper-large-v3 for Seed-TTS *test-en*, and Paraformer-zh for Seed-TTS *test-zh*, following previous works. For speech quality, we use Fréchet Speech Distance (FSD) with self-supervised wav2vec 2.0 [52] features, following [9]. For the subjective metrics, comparative mean option score (CMOS) and similarity mean option score (SMOS) are used to evaluate naturalness and similarity, respectively. CMOS is on a scale of -3 to 3, and SMOS is on a scale of 1 to 5.

**Baseline.** We compare our models with state-of-the-art zero-shot TTS systems, including Natural-Speech 3 [8], VALL-E [2], VoiceBox [9], VoiceCraft [5], XTTS-v2 [53], and CosyVoice [54]. More details of each model can be found in Appendix D. We also train an AR-based T2S model to replace the T2S part of MaskGCT, we term it as AR + SoundStorm.

**Training.** We train all models on 8 NVIDIA A100 80GB GPUs. We train two T2S models of different sizes (denoted as T2S-*Base* and T2S-*large*). For more details about the model architecture, please refer to Appendix A.1. We report the metrics of T2S-*large* by default, and you can find a comparison of model sizes in Section 4.5. We also compare two different methods of text tokenization: Grapheme-to-Phoneme (G2P) [55] and Byte Pair Encoding (BPE) [56]. See more details of the two methods in Appendix A.7. We report the metrics of G2P by default. We optimize these models with the AdamW [57] optimizer with a learning rate of 1e-4 and 32K warmup steps, following the inverse square root learning schedule. We use the classifier-free guidance [58], during training for both the T2S and S2A models, we drop the prompt with a probability of 0.15. See more details about classifier-free guidance and classifier-free guidance rescale in Appendix C.

**Inference.** For the T2S model, we use 50 steps as the default total inference steps. The classifier-free guidance scale and the classifier-free guidance rescale factor [59] are set to 2.5 and 0.75, respectively.

---

[2]`https://github.com/microsoft/UniSpeech/tree/main/downstreams/speaker_
verification`

[3]`https://huggingface.co/facebook/hubert-large-ls960-ft`

Table 1: Evaluation results for MaskGCT and the baseline methods on LibriSpeech *test-clean*, SeedTTS *test-en*, SeedTTS *test-zh*. The boldface denotes the best result, the underline denotes the second best. *gt length* denotes the result obtained by using ground truth total speech length. The results in '()' means the result is the best one selected from five random samples (rerank 5).

| System | SIM-O ↑ | WER ↓ | FSD ↓ | SMOS ↑ | CMOS ↑ |
|---|---|---|---|---|---|
| **LibriSpeech *test-clean*** | | | | | |
| Ground Truth | 0.68 | 1.94 | - | $4.05_{\pm 0.12}$ | 0.00 |
| VALL-E [2] | 0.50 | 5.90 | - | $3.47_{\pm 0.26}$ | $-0.52_{\pm 0.22}$ |
| VoiceBox [9] | 0.64 | 2.03 | 0.762 | $3.80_{\pm 0.17}$ | $-0.41_{\pm 0.13}$ |
| NaturalSpeech 3 [8] | 0.67 | **1.94** | 0.786 | $4.26_{\pm 0.10}$ | $\mathbf{0.16}_{\pm 0.14}$ |
| VoiceCraft [5] | 0.45 | 4.68 | 0.981 | $3.52_{\pm 0.21}$ | $-0.33_{\pm 0.16}$ |
| XTTS-v2 [53] | 0.51 | 4.20 | 0.945 | $3.02_{\pm 0.22}$ | $-0.98_{\pm 0.19}$ |
| MaskGCT | 0.687(0.723) | 2.634(1.976) | 0.886 | $\underline{4.27}_{\pm 0.14}$ | $0.10_{\pm 0.16}$ |
| MaskGCT (*rule-based*) | **0.686** | 2.976 | 0.797 | - | - |
| MaskGCT (*gt length*) | 0.697 | 2.012 | **0.746** | $\mathbf{4.33}_{\pm 0.11}$ | $\underline{0.13}_{\pm 0.13}$ |
| **SeedTTS *test-en*** | | | | | |
| Ground Truth | 0.730 | 2.143 | - | $3.92_{\pm 0.15}$ | 0.00 |
| CosyVoice [54] | 0.643 | 4.079 | 0.316 | $3.52_{\pm 0.17}$ | $-0.41_{\pm 0.18}$ |
| XTTS-v2 [53] | 0.463 | 3.248 | 0.484 | $3.15_{\pm 0.22}$ | $-0.86_{\pm 0.19}$ |
| VoiceCraft [5] | 0.470 | 7.556 | 0.226 | $3.18_{\pm 0.20}$ | $-1.08_{\pm 0.15}$ |
| MaskGCT | 0.717(0.760) | 2.623(1.283) | 0.188 | $\mathbf{4.24}_{\pm 0.12}$ | $\underline{0.03}_{\pm 0.14}$ |
| MaskGCT (*rule-based*) | 0.719 | 2.712 | 0.167 | - | - |
| MaskGCT (*gt length*) | **0.728** | **2.466** | **0.159** | $\underline{4.13}_{\pm 0.17}$ | $\mathbf{0.12}_{\pm 0.15}$ |
| **SeedTTS *test-zh*** | | | | | |
| Ground Truth | 0.750 | 1.254 | - | $3.86_{\pm 0.17}$ | 0.00 |
| CosyVoice [54] | 0.750 | 4.089 | 0.276 | $3.54_{\pm 0.12}$ | $-0.45_{\pm 0.15}$ |
| XTTS-v2 [53] | 0.635 | 2.876 | 0.413 | $2.95_{\pm 0.18}$ | $-0.81_{\pm 0.22}$ |
| MaskGCT | 0.774(0.805) | 2.273(0.843) | 0.106 | $\underline{4.09}_{\pm 0.12}$ | $\underline{0.05}_{\pm 0.17}$ |
| MaskGCT (*rule-based*) | 0.771 | 2.409 | 0.106 | - | - |
| MaskGCT (*gt length*) | **0.777** | **2.183** | **0.101** | $\mathbf{4.11}_{\pm 0.12}$ | $\mathbf{0.08}_{\pm 0.18}$ |

For sampling, we use a top-k of 20, with the sampling temperature annealing from 1.5 to 0. We add Gumbel noise to token confidences when determining the remasking process, following [11]. For the S2A model, we use $[40, 16, 1, 1, 1, 1, 1, 1, 1, 1, 1, 1]$ steps for acoustic RVQ layers by default, we find the S2A model can also perform well with fewer inference steps of $[10, 1, 1, 1, 1, 1, 1, 1, 1, 1, 1, 1]$ (see Appendix A.3). We use the same sampling strategy as the T2S model, except that we use greedy sampling instead of top-k sampling if the inference step is 1.

## 4.2 ZERO-SHOT TTS

In this section, we show the main results of zero-shot TTS: we show comparison results with SOTA baselines in Section 4.2.1; we compare MaskGCT with replacing T2S model to an AR model in Section 4.2.2; We present the performance of MaskGCT across varying speech tempos in Section 4.2.3. Additionally, we present the results of zero-shot TTS for speech style imitation in Section 4.3, multilingual zero-shot TTS in Appendix E, and cross-lingual speech translation (dubbing) in Appendix F.

### 4.2.1 COMPARISON WITH BASELINES

We compare MaskGCT with baselines in terms of similarity, robustness, and generation quality. The main results are shown in Table 1. MaskGCT demonstrates excellent performance on all metrics and achieves human-level similarity, naturalness, and intelligibility. In *similarity*, MaskGCT's SIM-O and SMOS both outperform the best baseline, whether assessed using the total length of ground truth or the predicted total duration (0.67→0.687 in LibriSpeech, 0.643→0.717 in SeedTTS *test-en*, 0.75→0.774 in SeedTTS *test-zh* for SIM-O; +0.01 in LibriSpeech, +0.72 in SeedTTS *test-en*, +0.55 in SeedTTS *test-zh* for SMOS). When compared with human recordings, MaskGCT achieves human-level similarity across all three test sets (+0.017, -0.002, and +0.027 for SIM-O respectively in the three test sets, and +0.28, +0.32, and +0.25 for SMOS respectively in the three test sets). In *robustness*, MaskGCT likewise results nearly on par with ground truth (with 2.634, 2.623, 2.273 WER on LibriSpeech, SeedTTS *test-en*, and SeedTTS *test-zh*, respectively), exhibiting enhanced

robustness compared to AR-based models and performing on par or better than NAR-based models such as VoiceBox and NaturalSpeech 3, without relying on phone-level duration predictions. In *generation quality*, MaskGCT achieves +0.10, +0.03, and +0.05 CMOS across the three test sets when compared with human recordings, indicating that MaskGCT attains human-level naturalness on these test sets. We also observe that MaskGCT exhibits excellent performance when using both ground truth total duration and predicted total duration, indicating the robustness of MaskGCT within a reasonable range of total speech duration.

Table 2: Comparison results of the evaluation of MaskGCT and AR+SoundStorm. AR+SoundStorm can be regarded as replacing the T2S MaskGCT with the AR T2S model.

| System | SIM-O ↑ | WER ↓ | FSD ↓ | SMOS ↑ | CMOS ↑ |
|---|---|---|---|---|---|
| **LibriSpeech *test-clean*** | | | | | |
| AR + SoundStorm | 0.672 | 3.267 | 0.998 | $4.20_{\pm 0.17}$ | $-0.02_{\pm 0.20}$ |
| MaskGCT | **0.687** | **2.634** | **0.886** | $\mathbf{4.27}_{\pm 0.14}$ | $\mathbf{0.10}_{\pm 0.16}$ |
| **SeedTTS *test-en*** | | | | | |
| AR + SoundStorm | 0.683 | 2.846 | 0.323 | $4.03_{\pm 0.23}$ | $-0.05_{\pm 0.22}$ |
| MaskGCT | **0.717** | **2.623** | **0.188** | $\mathbf{4.24}_{\pm 0.12}$ | $\mathbf{0.03}_{\pm 0.14}$ |
| **SeedTTS *test-zh*** | | | | | |
| AR + SoundStorm | 0.747 | 3.865 | 0.238 | $3.78_{\pm 0.23}$ | $-0.32_{\pm 0.19}$ |
| MaskGCT | **0.774** | **2.273** | **0.106** | $\mathbf{4.09}_{\pm 0.12}$ | $\mathbf{0.05}_{\pm 0.17}$ |

### 4.2.2 AUTOREGRESSIVE VS. MASKED GENERATIVE MODELS

We compare MaskGCT to replacing T2S MaskGCT with an AR T2S model (which we call AR + SoundStorm). Table 2 shows the performance of these two models on all three test sets. MaskGCT demonstrates improved similarity, robustness, and CMOS (+0.12 on LibriSpeech *test-clean*, +0.08 on SeedTTS *test-en*, and +0.37 on SeedTTS *test-zh*) across all three test sets. We also conduct comparisons on more challenging hard cases (such as repeating words, and tongue twisters, which are often considered as samples where TTS systems are prone to **hallucinations**). MaskGCT exhibits a more pronounced robustness advantage in these scenarios. See details in Appendix J. In addition, compared to AR-based models, MaskGCT offers the capability to control the total duration of the generated speech, along with fewer inference steps, requiring only 25 to 50 steps for T2S models to achieve optimal results for speeches of any length. Conversely, the inference steps for AR-based models increase linearly with the length of the speech.

### 4.2.3 DURATION LENGTH ANALYSIS

We analyze the robustness of the generated results of MaskGCT under different changes in total duration length (which can also be regarded as changes in speech tempo). The results are shown in Figure 3. We explore the results of multiplying the ground truth total duration by 0.7 to 1.3. The results show that the lowest WER is achieved at a total duration multiplier of 1.0, indicating that the models perform best when the speech is played at its natural speed. When

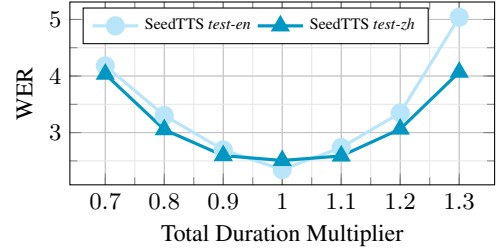

Figure 3: WER vs. Total Duration Multiplier.

the multiplier is 0.9 or 1.1, the model is still able to achieve a WER very close to the best. When the multiplier is 0.7 or 1.3, the WER is slightly higher but still within a reasonable range. This shows that our model can generate reasonable and accurate content at different speech tempos.

### 4.3 SPEECH STYLE IMITATION

Zero-shot TTS endeavors to learn **how to speak**, including voice timbre and style, from prompt speech. Previous works utilized SIM-O to measure the similarity between generated speech and reference speech; however, SIM-O primarily assesses the similarity in voice timbre. In addition to evaluating the model's zero-shot cloning ability through timbre similarity metrics, we also explored MaskGCT's capability to clone overall style from two more expressive and stylized dimensions: accent and

emotion. We randomly sampled a portion of data from the L2-ARCTIC [60] accent corpus and the ESD [61] emotion corpus to construct our accent and emotion evaluation datasets. Additionally, we introduce supplementary metrics to assess the model's performance. For accent imitation, we employ SIM-Accent, to measure the similarity in accent between the generated speech and reference speech. The calculation process is analogous to SIM-O, but we utilize CommonAccent[4] [62, 63] to derive the accent representation features of the speech. We also incorporate a subjective evaluation metric, Accent SMOS, which is similar to SMOS but focuses on accent rather than timbre. For emotion, we introduce Emotion SIM (with emotion2vec[5] [64] to extract features) and Emotion SMOS.

Our experiments demonstrate that MaskGCT exhibits powerful style cloning capabilities. For accent imitation, MaskGCT achieves the highest SIM-O of 0.717, close to the ground truth of 0.747. It also maintains a competitive WER of 6.382 and the best Accent SIM of 0.645. Additionally, MaskGCT leads in CMOS of 0.23, SMOS of 4.24, and Accent SMOS of 4.38. For emotion imitation, MaskGCT achieves the highest SIM-O of 0.600. It also attains a competitive WER of 12.502 and a strong Emotion SIM of 0.822. Furthermore, MaskGCT leads in all subjective metrics with CMOS of -0.31, SMOS of 4.07, and Emotion SMOS of 3.76, indicating natural and pleasant emotion imitation.

Table 3: Evaluation results for MaskGCT and the baseline methods on accent imitation.

| System | SIM-O ↑ | WER ↓ | Accent SIM ↑ | CMOS ↑ | SMOS ↑ | Accent SMOS ↑ |
|---|---|---|---|---|---|---|
| **Accent Corpus *L2-Arctit*** | | | | | | |
| Ground Truth | 0.747 | 10.903 | 0.633 | 0.00 | - | - |
| VALL-E | 0.403 | 10.721 | 0.485 | $-1.04_{\pm 0.50}$ | $3.12_{\pm 0.41}$ | $2.77_{\pm 0.45}$ |
| CosyVoice | 0.653 | 6.660 | 0.640 | $0.10_{\pm 0.19}$ | $4.23_{\pm 0.18}$ | $3.99_{\pm 0.23}$ |
| VoiceBox | 0.475 | **6.181** | 0.575 | $-0.55_{\pm 0.22}$ | $3.93_{\pm 0.25}$ | $3.49_{\pm 0.29}$ |
| VoiceCraft | 0.438 | 10.072 | 0.517 | $-0.39_{\pm 0.22}$ | $3.51_{\pm 0.33}$ | $3.29_{\pm 0.28}$ |
| MaskGCT | **0.717** | 6.382 | **0.645** | $\mathbf{0.23}_{\pm 0.17}$ | $\mathbf{4.24}_{\pm 0.16}$ | $\mathbf{4.38}_{\pm 0.25}$ |

Table 4: Evaluation results for MaskGCT and the baseline methods on emotion imitation.

| System | SIM-O ↑ | WER ↓ | Emotion SIM ↑ | CMOS ↑ | SMOS ↑ | Emotion SMOS ↑ |
|---|---|---|---|---|---|---|
| **Emotion Corpus *ESD*** | | | | | | |
| Ground Truth | 0.673 | 11.792 | 0.936 | 0.00 | - | - |
| VALL-E | 0.396 | 15.731 | 0.735 | $-1.43_{\pm 0.33}$ | $2.52_{\pm 0.38}$ | $2.63_{\pm 0.36}$ |
| CosyVoice | 0.575 | **10.139** | **0.839** | $-0.45_{\pm 0.18}$ | $3.98_{\pm 0.19}$ | $3.66_{\pm 0.19}$ |
| VoiceBox | 0.451 | 12.647 | 0.811 | $-0.65_{\pm 0.20}$ | $3.81_{\pm 0.16}$ | $3.61_{\pm 0.19}$ |
| VoiceCraft | 0.345 | 16.042 | 0.788 | $-0.60_{\pm 0.24}$ | $3.42_{\pm 0.31}$ | $3.52_{\pm 0.25}$ |
| MaskGCT | **0.600** | 12.502 | 0.822 | $\mathbf{-0.31}_{\pm 0.17}$ | $\mathbf{4.07}_{\pm 0.16}$ | $\mathbf{3.76}_{\pm 0.25}$ |

## 4.4 Choice of Semantic Representation Codec

In this section, we investigate the impact of different semantic representation approaches on acoustic token reconstruction. We primarily evaluate two types of semantic codecs: VQ-based and k-means-based. For VQ-based approaches, we implement two configurations with codebook sizes of 8192 and 2048, denoted as "VQ 8192" and "VQ 2048", respectively. Similarly, for k-means-based approaches, we train models with 8192 and 2048 clusters, denoted as "k-means 8192" and "k-means 2048". To assess how different semantic representations affect acoustic prediction, we train separate semantic-to-acoustic (S2A) models for each configuration and evaluate their performance through speech reconstruction metrics. The comparative results are presented in Table 5.

Table 5: Evaluation results for S2A models with different semantic codecs. In this experiment, we use the ground-truth semantic tokens to predict acoustic tokens.

| Semantic Codec | SIM-O ↑ | WER ↓ | SIM-O ↑ | WER ↓ | SIM-O ↑ | WER ↓ |
|---|---|---|---|---|---|---|
| | **LibriSpeech *test-clean*** | | **SeedTTS *test-en*** | | **SeedTTS *test-zh*** | |
| k-means 2048 | 0.648 | 3.013 | 0.658 | 3.989 | 0.691 | 11.420 |
| k-means 8192 | 0.661 | 2.862 | 0.664 | 3.012 | 0.713 | 8.782 |
| VQ 2048 | 0.671 | **2.177** | 0.692 | 3.187 | 0.744 | 4.913 |
| VQ 8192 | **0.680** | 2.223 | **0.713** | **2.175** | **0.763** | **2.088** |

[4]https://huggingface.co/Jzuluaga/accent-id-commonaccent_ecapa
[5]https://github.com/ddlBoJack/emotion2vec

The experimental results demonstrate consistent patterns across all three test sets. For SIM-O, the results reveal a consistent performance ranking across all test sets, with VQ 8192 yielding the highest scores, followed sequentially by VQ 2048, k-means 8192, and k-means 2048. Regarding Word Error Rate (WER), VQ-based methods consistently outperform k-means approaches, though this advantage is less pronounced in the LibriSpeech *test-clean* set. Notably, on the SeedTTS *test-zh* (Chinese test set), k-means exhibits a substantial degradation in the performance of the WER. We attribute this phenomenon to the stronger coupling between semantic and prosodic features in Chinese, where the transition from VQ to k-means results in a significant loss of prosodic information in the semantic representations.

## 4.5 ABLATION STUDY

**Inference Timesteps.** We explore the impact of inference steps of the T2S model on the results, ranging from 5 steps to 75 steps. Initially, SIM increases significantly and stabilizes after 25 steps. For *test-zh*, it rises from 0.761 at 5 steps to 0.771 at 75 steps, and for *test-en*, from 0.696 to 0.715. SIM peaks around 25 steps. WER improves more dramatically, especially up to 25 steps. For *test-zh*, it drops from 10.19 at 5 steps to 2.507 at 25 steps, and for *test-en*, from 8.096 to 2.346. Both SIM and WER show minimal changes beyond 25 steps. These findings suggest that SIM can be optimized with around 10 steps, while achieving the lowest WER requires approximately 25 steps. Beyond this, both metrics show minimal changes, indicating that further increases in steps do not yield substantial improvements. For practical applications, 25 inference steps may be optimal for balancing SIM and WER, ensuring efficient and effective performance. See more details in Appendix A.2.

**Model Size.** We compare the performance differences of T2S models with varying model sizes. The result is shown in Table 6. We observe that the large model outperforms the base model across all metrics, albeit not significantly. We suggest that our system can achieve good performance with just the setting of the base model when using 100K hours of data. In the future, we will explore more comprehensive scaling laws for both model size and data scaling.

Table 6: Comparison results between T2S-*Large* and T2S-*Base*.

| System | SIM-O ↑ | WER ↓ | FSD ↓ | #Parameters |
|---|---|---|---|---|
| **SeedTTS *test-en*** | | | | |
| T2S-*Base* | 0.714 | 2.514 | 0.189 | 315M |
| T2S-*Large* | **0.728** | **2.466** | **0.159** | 695M |
| **SeedTTS *test-zh*** | | | | |
| T2S-*Base* | 0.769 | 2.216 | 0.123 | 315M |
| T2S-*Large* | **0.777** | **2.183** | **0.101** | 695M |

**Text Tokenizer.** We compare two text tokenization methods: Grapheme-to-Phoneme (G2P) and Byte Pair Encoding (BPE). See more details in Appendix A.7.

## 5 CONCLUSION

In this paper, we present MaskGCT, a large-scale zero-shot TTS system that leverages fully non-autoregressive masked generative codec transformers while not requiring text-speech alignment supervision and phone-level duration prediction. MaskGCT achieves high-quality text-to-speech synthesis using text to predict semantic tokens extracted from a speech self-supervised learning (SSL) model, and then predicting acoustic tokens conditioned on these semantic tokens. Our experiments demonstrate that MaskGCT outperforms the state-of-the-art TTS system on speech quality, similarity, and intelligibility with scaled model size and training data, and MaskGCT can control the total duration of generated speech. We also explore the scalability of MaskGCT in tasks such as speech translation, voice conversion, emotion control, and speech content editing, demonstrating the potential of MaskGCT as a foundational model for speech generation.

## ACKNOWLEDGMENT

This work is partially supported by the NSFC under Grant 62376237, Shenzhen Science and Technology Program ZDSYS20230626091302006. We thank all anonymous reviewers for their insightful comments and suggestions.

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

# A    DETAILS OF MASKGCT

## A.1    MODEL ARCHITECTURE

We use a Llama-style [42] Transformer architecture as the backbone of our model, incorporating gated linear units with GELU [43] activation (SwiGLU), rotation position encoding [44], etc., but replacing causal attention with bidirectional attention. We also use adaptive RMSNorm [45], which accepts the time step $t$ as the condition. Table 7 presents the key hyperparameters of the models.

Table 7: Overview of the key hyperparameters of MaskGCT.

|  | **T2S-*Base*** | **T2S-*Large*** | **S2A** |
|---|---|---|---|
| Layers | 16 | 16 | 16 |
| Model Dimension | 1,024 | 1,536 | 1,024 |
| FFN Dimension | 4,096 | 6,144 | 4,096 |
| Attention Heads | 16 | 16 | 16 |
| Attention Type | Bidirectional | Bidirectional | Bidirectional |
| Activation Function | SwiGLU | - | - |
| Positional Embeddings | RoPE ($\theta$ = 10,000) | - | - |
| Number of Parameters | 315M | 695M | 353M |

## A.2    INFERENCE STEPS FOR THE T2S MODEL

Figure 4 shows the relationship between inference steps and metrics SIM and WER for SeedTTS *test-zh* (left) and *test-en* (right). Initially, SIM increases significantly, stabilizing after 25 steps. For *test-zh*, SIM rises from 0.761 at 5 steps to 0.771 at 75 steps, and for *test-en*, from 0.696 to 0.715. SIM reaches high values with just 10 steps but peaks around 25 steps. WER improves more dramatically, especially up to 25 steps. For *test-zh*, WER drops from 10.19 at 5 steps to 2.507 at 25 steps, and for *test-en*, from 8.096 to 2.346. Both SIM and WER show minimal changes beyond 25 steps. These findings indicate that while SIM metrics can be sufficiently optimized with around 10 inference steps, achieving the lowest WER values requires approximately 25 inference steps. Beyond this threshold, both SIM and WER metrics exhibit minimal changes, implying that further increases in inference steps do not yield substantial improvements in these performance metrics. Therefore, for practical applications, 25 inference steps may be considered optimal for balancing SIM and WER, ensuring efficient and effective performance.

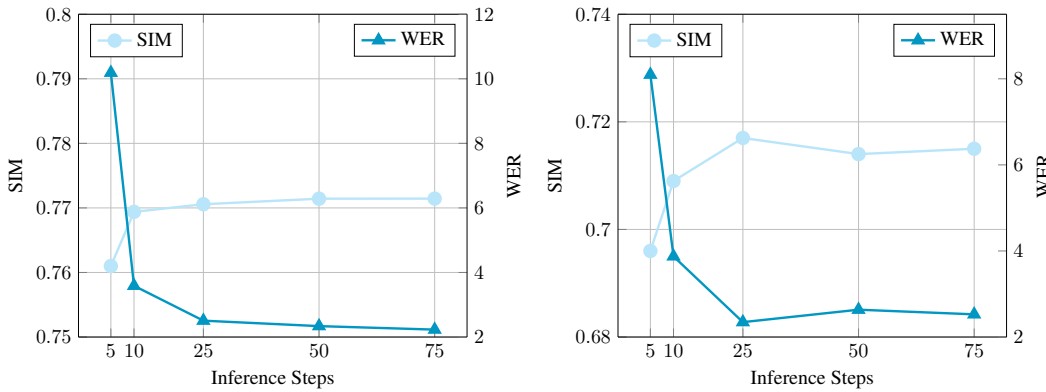

Figure 4: Inference Steps vs. SIM and WER. The results on the left are for SeedTTS *test-zh*, and the results on the right are for SeedTTS *test-en*. In this ablation study, we utilize the ground truth speech length.

A.3 INFERENCE STEPS FOR THE S2A MODEL

The S2A model generates tokens layer by layer during inference. Since the acoustic codec follows an RVQ structure, we can view the S2A inference as a process from coarse to fine. We also use more iterations in the initial layers, as the first few layers carry more information. By default, we use inference steps of $[40, 16, 1, 1, 1, 1, 1, 1, 1, 1, 1, 1]$ for each layer, however, we find that the S2A model can also perform well with fewer steps, such as $[10, 1, 1, 1, 1, 1, 1, 1, 1, 1, 1, 1]$, with only a very slight performance loss.

Table 8: Evaluation results of different inference steps for the S2A model.

| Inference Steps | SIM-O ↑ | WER ↓ | FSD ↓ |
|---|---|---|---|
| SeedTTS *test-en* | | | |
| $[10, 1, 1, 1, 1, 1, 1, 1, 1, 1, 1, 1]$ | 0.709 | 2.796 | 0.164 |
| $[40, 16, 1, 1, 1, 1, 1, 1, 1, 1, 1, 1]$ | **0.728** | **2.466** | **0.159** |
| SeedTTS *test-zh* | | | |
| $[10, 1, 1, 1, 1, 1, 1, 1, 1, 1, 1, 1]$ | 0.766 | 2.268 | 0.111 |
| $[40, 16, 1, 1, 1, 1, 1, 1, 1, 1, 1, 1]$ | **0.777** | **2.183** | **0.101** |

We present the real-time factor (RTF) of MaskGCT on an A100 GPU for generating a 20-second speech across various inference steps in Table 9. Across all configurations presented, there is no significant performance difference. Additionally, we also present the RTF of AR + SoundStorm. For AR + SoundStorm, generating a 20-second speech requires 1000 steps for text-to-semantic inference. However, it can leverage kv-cache to accelerate the process.

Table 9: Real-time factor (RTF) comparison of MaskGCT and AR + SoundStorm on an A100 GPU for generating a 20-second speech.

| Model | T2S steps | S2A steps | RTF |
|---|---|---|---|
| MaskGCT | 50 | $[40, 16, 1, 1, 1, 1, 1, 1, 1, 1, 1, 1]$ | 0.52 |
| MaskGCT | 50 | $[10, 1, 1, 1, 1, 1, 1, 1, 1, 1, 1, 1]$ | 0.44 |
| MaskGCT | 25 | $[10, 1, 1, 1, 1, 1, 1, 1, 1, 1, 1, 1]$ | 0.31 |
| AR + SoundStorm | 1000 | $[40, 16, 1, 1, 1, 1, 1, 1, 1, 1, 1, 1]$ | 0.98 |

A.4 DETAILS OF SEMANTIC AND ACOUSTIC CODEC

For semantic codec, we train a VQ-VAE model using the hidden features from the 17th layer of W2v-BERT 2.0, incorporating factorized codec [33] technology. The original hidden dimension of 1,024 is projected into a lower-dimensional space for quantization. The codebook size is set to 8,192, with a codebook dimension of 8. We employ only the $\mathcal{L}_1$ loss as the reconstruction target, optimizing the codebook with codebook loss and commitment loss. The input features are normalized to have a mean of 0 and a variance of 1, based on the statistics of the training dataset. The encoder and the decoder are each composed of 12 mirrored ConvNext blocks, featuring a kernel size of 7 and a hidden size of 384.

For acoustic codec, the basic architecture of the encoder follows [37] and the decoder follows [47]. The Vocos-based decoder can model amplitude and phase, enabling waveform generation through inverse STFT transformation without requiring upsampling. The number of RVQ layers, codebook size, and codebook dimension are set to 12, 8,192, and 8, respectively. We utilize the multi-scale mel-reconstruction loss [37] $\mathcal{L}_{\text{rec}}$, for the adversarial loss $\mathcal{L}_{\text{adv}}$, we employ both the multi-period discriminator (MPD) and the multi-band multi-scale STFT discriminator, as proposed by [37, 65]. Additionally, we incorporate the relative feature matching loss $\mathcal{L}_{\text{feat}}$. For codebook learning, we use the codebook loss $\mathcal{L}_{\text{codebook}}$ and the commitment loss $\mathcal{L}_{\text{commit}}$ from VQ-VAE. We set $\lambda_{\text{rec}} = 10.0$, $\lambda_{\text{adv}} = 2.0$, $\lambda_{\text{feat}} = 2.0$, $\lambda_{\text{codebook}} = 1.0$, $\lambda_{\text{commit}} = 0.25$ as coefficients for balancing each loss terms. Figure 5 shows the overview of the semantic codec and acoustic codec, Table 10 presents the detailed model configurations of semantic codec and acoustic codec.

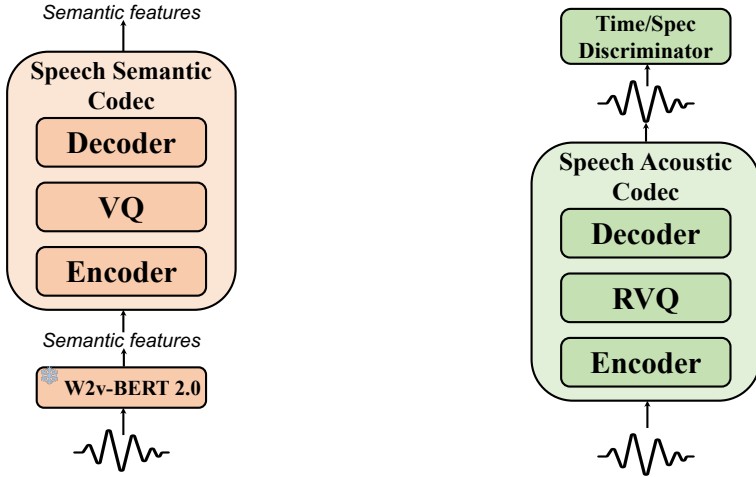

Figure 5: An overview of the semantic codec (left) and acoustic codec (right). The semantic codec is trained to quantize semantic features with a single codebook and reconstruct semantic features. The acoustic codec is trained to quantize and reconstruct the speech waveform using RVQ, with time and spectral discriminators to enhance the reconstruction quality further.

Table 10: The detailed model configurations of semantic codec and acoustic codec.

|  | Semantic Codec | Acoustic Codec |
| --- | --- | --- |
| Input | W2v-BERT 2.0 hidden | Waveform |
| Sample Rate | 16K | 24K |
| Hopsize | 320 | 480 |
| Number of (R)VQ Blocks | 1 | 12 |
| Codebook size | 8,192 | 1,024 |
| Codebook Dimension | 8 | 8 |
| Decoder Hidden Dimension | 384 | 512 |
| Decoder Kernel Size | 7 | 7 |
| Number of Decoder Blocks | 12 | 30 |
| Number of Parameters | 44M | 170M |

A.5    DETAILS OF DURATION PREDICTOR

MaskGCT requires specifying the target speech duration during inference, so we train a flow matching [46, 66] based duration predictor to obtain the total duration of the target audio by summing the phone-level duration. Note that we do not need to actually use the phone-level durations but only use them to make a reasonable estimate of the total duration, leaving other total duration predictor methods for future works to explore. The duration predictor has a similar Transformer architecture to MaskGCT, with 12 layers, 12 attention heads, and a hidden size of 768. We also adapt in-context learning and classifier-free guidance for the duration predictor. During training, we randomly select a prefix segment of the phoneme sequence and its corresponding duration as a prompt, which is not added with noise. At the same time, we use a probability of 0.15 to drop the prompt. We model the duration in the log domain using flow matching. We denote $x_1$ as a random variable of $\log(\text{duration}+1)$, $x_0$ as a randomly sampled Gaussian noise, then $v_\theta(x_t, t) = x_t = (1-t)x_0 + tx_1$, where the timestep $t \in [0, 1]$. The loss function of the duration predictor is $\mathbb{E}_{t,x_1}(v_\theta(x_t, t) - (x_1 - x_0))^2$. In the inference stage, we use a midpoint ODE solver to generate the target from randomly sampled Gaussian noise with a total of 4 steps. We pretrain a duration aligner (between phoneme and W2v-BERT 2.0 semantic feature) based on monotonic alignment search (MAS) [67] to get the ground truth duration for each phoneme.

It is noteworthy that there are several methods to determine the total duration length. Our trained duration predictor is solely for providing a rough estimate to facilitate inference and comparison. Alterna-

tively, a simpler approach could be: *target total duration = target phone number* $\times \frac{\textit{prompt total duration}}{\textit{prompt phone number}}$.
Table 11 illustrates the comparative results of MaskGCT under three different total duration calculation methods. The results indicate that our model, using simple rules to predict total duration, can generate speech with SIM and WER that are essentially comparable to those of the ground truth. The results indicate that our model, using simple rules to predict total duration, can generate speech with SIM and WER that are essentially comparable to those of the ground truth.

Table 11: Comparison of results for MaskGCT under three different total duration calculation methods.

| Method | SIM-O ↑ | WER ↓ |
|---|---|---|
| **LibriSpeech *test-clean*** | | |
| *rule-based* | 0.686 | 2.976 |
| *duration predictor* | 0.687 | 2.634 |
| *gt length* | 0.697 | 2.012 |
| **SeedTTS *test-en*** | | |
| *rule-based* | 0.719 | 2.712 |
| *duration predictor* | 0.717 | 2.623 |
| *gt length* | 0.728 | 2.466 |
| **SeedTTS *test-zh*** | | |
| *rule-based* | 0.771 | 2.409 |
| *duration predictor* | 0.774 | 2.273 |
| *gt length* | 0.777 | 2.183 |

## A.6 COMPARISON BETWEEN MASKGCT AND OTHER SYSTEMS

Table 12: A comparison between MaskGCT and existing systems. "Model" stands for modeling method and "Rep." stands for the representation used. MaskGCT uses masked generative modeling for acoustic and semantic tokens ("A." stands for acoustic, "S." stands for semantic, "F." stands for factorized tokens used in NaturalSpeech 3). MaskGCT implicitly models duration ("Imp. Dur.") and allows flexible control over the total length of generated speech ("Len. Ctrl"). MaskGCT supports various speech generation tasks. "ZS TTS" denotes zero-shot TTS and "CL TTS" denotes cross-lingual TTS.

| System | Model | Rep. | Imp. Dur. | Len. Ctrl. | ZS TTS | CL TTS | Dubbing | Edit |
|---|---|---|---|---|---|---|---|---|
| VALL-E | Autoregressive | A. Tokens | ✓ | ✗ | ✓ | ✗ | ✗ | ✗ |
| NaturalSpeech 2 | Diffusion | A. Features | ✗ | ✗ | ✓ | ✗ | ✗ | ✗ |
| VoiceBox | Diffusion | A. Features | ✗ | ✗ | ✓ | ✓ | ✗ | ✓ |
| VoiceCraft | Autoregressive | A. Tokens | ✓ | ✗ | ✓ | ✗ | ✗ | ✓ |
| NaturalSpeech 3 | Masked Generative | F. Tokens | ✗ | ✗ | ✓ | ✗ | ✗ | ✓ |
| MaskGCT | Masked Generative | S.&A. Tokens | ✓ | ✓ | ✓ | ✓ | ✓ | ✓ |

## A.7 TEXT TOKENIZER

We consider two text tokenization methods: Grapheme-to-Phoneme (G2P) and Byte Pair Encoding (BPE). For G2P, we employ phonemize[6] for English and a combination of jieba[7] and pypinyin[8] for Chinese. For BPE, we utilize the BPE method and vocabulary from Whisper[9], with a vocabulary size exceeding 30,000. Table 13 shows the comparison results of MaskGCT using the two different text tokenization methods. The results indicate that G2P outperforms BPE in English with a higher SIM-O of 0.728 compared to 0.711 and a lower WER of 2.466 versus 4.036. Conversely, in Chinese, G2P maintains a slightly higher SIM-O (0.777 vs. 0.769) but BPE achieves a lower WER (1.921 vs.

Table 13: G2P vs. BPE.

| | SIM-O ↑ | WER ↓ |
|---|---|---|
| **SeedTTS *test-en*** | | |
| G2P | **0.728** | **2.466** |
| BPE | 0.711 | 4.036 |
| **SeedTTS *test-zh*** | | |
| G2P | **0.777** | 2.183 |
| BPE | 0.769 | **1.921** |

---

[6] https://github.com/bootphon/phonemizer
[7] https://github.com/fxsjy/jieba
[8] https://github.com/mozillazg/python-pinyin
[9] https://github.com/huggingface/transformers/blob/main/src/transformers/models/whisper/tokenization_whisper.py

2.338). These findings suggest that while G2P is superior in preserving text similarity and reducing errors in English, BPE is more effective in minimizing WER in Chinese. We hypothesize that the reason might be that the Chinese G2P system we used still has deficiencies in handling polyphonic characters. In contrast, BPE can learn different pronunciations for the same character based on context.

## B DISCUSSION ABOUT SEMANTIC AND ACOUSTIC DEFINITIONS

In this paper, we refer to the speech representation extracted from the speech self-supervised learning (SSL) model as the semantic feature. The discrete tokens obtained through the discretization of these semantic features (using k-means or vector quantization are termed semantic tokens. Similarly, we define the representations from melspectrogram, neural speech codecs, or speech VAE as acoustic features, and their discrete counterparts are called acoustic tokens. This terminology was first introduced in [68] and has since been adopted by many subsequent works [8, 19, 39, 69, 70]. It is important to note that this is not a strictly rigorous definition. Generally, we consider semantic features or tokens to contain more prominent linguistic information and exhibit stronger correlations with phonemes or text. One measure of this is the phonetic discriminability in terms of the ABX error rate. In this paper, the W2v-BERT 2.0 features we use have a phonetic discriminability within less than 5 on the LibriSpeech *dev-clean* dataset, whereas acoustic features, for example, Encodec latent features, score above 20 on this metric. However, it is worth noting that semantic features or tokens not only contain semantic information but also include prosodic and timbre aspects. In fact, we suggest that for certain two-stage zero-shot TTS systems, excessive loss of information in semantic tokens can degrade the performance of the second stage, where semantic-to-acoustic conversion occurs. Therefore, finding a speech representation that is more suitable for speech generation remains a challenging problem.

## C CLASSIFIER-FREE GUIDANCE

We adopt the classifier-free guidance [58] technique for both the T2S model and the S2A model. We also introduce classifier-free guidance with rescaling, following [59]. In the training stage, we randomly drop the prompt with a probability of 0.15 to model the probability distribution $p_\theta(\mathbf{X})$ without the prompt. During inference, we compute the output embedding $g_\theta^{\mathrm{cfg}}(\mathbf{X}|\mathbf{X}^p) = g_\theta(\mathbf{X}|\mathbf{X}^p) + w_{\mathrm{cfg}} \cdot (g_\theta(\mathbf{X}|\mathbf{X}^p) - g_\theta(\mathbf{X}))$ of the last layer of the model, where $w_{\mathrm{cfg}}$ is the classifier-free guidance scale, then we compute the rescale embedding $g_\theta^{\mathrm{rescale}}(\mathbf{X}|\mathbf{X}^p) = g_\theta^{\mathrm{cfg}}(\mathbf{X}|\mathbf{X}^p) \times \mathrm{std}(g_\theta(\mathbf{X}|\mathbf{X}^p))/\mathrm{std}(g_\theta^{\mathrm{cfg}}(\mathbf{X}|\mathbf{X}^p))$, the final output embedding is computed as $w_{\mathrm{rescale}} \cdot g_\theta^{\mathrm{rescale}}(\mathbf{X}|\mathbf{X}^p) + (1 - w_{\mathrm{rescale}}) \cdot g_\theta^{\mathrm{cfg}}(\mathbf{X}|\mathbf{X}^p)$. In our paper, $w_{\mathrm{cfg}}$ and $w_{\mathrm{rescale}}$ are set as 2.5 and 0.75 by default.

## D EVALUATION BASELINES

VALL-E [2]. A large-scale TTS system uses an autoregressive and an additional non-autoregressive model to predict discrete tokens from a neural speech codec [26]. We reproduce VALL-E with Amphion toolkit [71] and Librilight [72] dataset.

NaturalSpeech 3 [8]. A non-autoregressive model large-scale TTS systems with factorized speech codec for speech decoupling representation and factorized diffusion Models for speech generation. It achieves human-level naturalness on the LibriSpeech test set. We report the scores of LibriSpeech *test-clean* obtained from [8] and ask for the generated samples for subjective evaluation.

VoiceBox [9]. A non-autoregressive model large-scale multi-task speech generation model based on flow matching [46]. We report the scores of LibriSpeech *test-clean* obtained from [8] and ask for the generated samples for subjective evaluation.

XTTS-v2 [53]. An open-source multilingual TTS model that supports 16 languages. It is also based on an autoregressive model. We use the official code and pre-trained checkpoint[10].

---

[10]`https://huggingface.co/coqui/XTTS-v2`

VoiceCraft [5]. A token-infilling neural codec language model for text editing and text-to-speech. It predicts multi-layer tokens in a delay pattern. We use the official code and pre-trained checkpoint[11].

CosyVoice [54]. A two-stage large-scale TTS system. The first stage is an autoregressive model and the second stage is a diffusion model. It is trained on 170,000 hours of multilingual speech data. We use the official code and pre-trained checkpoint[12].

# E    MULTILINGUAL ZERO-SHOT TTS

We validate the effectiveness of MaskGCT across four additional languages beyond Chinese and English, specifically Japanese, Korean, German, and French. On the foundation of our existing training data, we expand by 2,500 hours of Japanese, 7,400 hours of Korean, 6,900 hours of German, and 8,200 hours of French. We collect these data using the data collection pipeline proposed by [48]. For evaluation, we use the test sets provided in [48]. We still employ SIM-O and WER as evaluation metrics, with Whisper-medium[13] serving as the ASR model for WER assessment. We utilize XTTS-v2 and the two models proposed in [48]: Emilia-AR and Emilia-NAR as comparative baselines. Table 14 shows the results. MaskGCT demonstrates significant improvements over the baselines, with the exception of WER in Japanese. It is noteworthy that we only retrained our text-to-semantic model using the expanded data, without retraining the tokenizers and semantic-to-acoustic models. We believe that further enhancements in our model's performance can be achieved if all components are retrained on the expanded data.

Table 14: Evaluation results for MaskGCT and baseline methods on the test sets for Japanese, Korean, German, and French.

| System | Ja | | Ko | | Fr | | De | |
|---|---|---|---|---|---|---|---|---|
| | WER | SIM-O | WER | SIM-O | WER | SIM-O | WER | SIM-O |
| Emilia-AR | 3.6 | 0.625 | 10.9 | 0.681 | 8.2 | 0.589 | 6.8 | 0.680 |
| Emilia-NAR | 10.8 | 0.562 | 15.2 | 0.608 | 17.5 | 0.550 | 13.3 | 0.633 |
| XTTS-v2 | **2.981** | 0.579 | 12.45 | 0.617 | 6.898 | 0.531 | 9.168 | 0.569 |
| MaskGCT | 3.903 | **0.678** | **9.417** | **0.732** | **5.598** | **0.667** | **5.126** | **0.745** |

# F    DURATION-CONTROLLABLE SPEECH TRANSLATION

The goal of the speech translation task is to translate speech from one language to another while preserving the original semantic, timbre, and prosody. In some scenarios, we also need to ensure that the total duration remains relatively unchanged, such as in cross-lingual dubbing. Our model can achieve this seamlessly, with the ability to control the total duration and, through in-context learning, use the pre-translation speech as a prompt to maintain the timbre and prosody. To quantify the capabilities of our model, we randomly select 200 samples from SeedTTS *test-zh* and 200 samples from SeedTTS *test-en*. Additionally, we sample 200 examples for each language of Japanese, Korean, German, and French from each of the test sets provided in [48]. Subsequently, we utilize GPT4o-mini [73] to translate each sample into one of the other five languages, using the translated text as the target text. We use the duration of prompt speech as the duration of target speech. This process yields 30 sets of test data. Table 15 shows the results of the 30 sets of experiments. We observe that MaskGCT maintains a good level of speaker similarity across translations between the six languages. Both "X to En" and "En to X" generally perform well, characterized by relatively low WER values and moderate SIM-O scores. "X to Ja" also achieve low WER values. However, for languages other than English, "X to Zh", "X to De", and "X to Fr" exhibit higher WER values. We hypothesize that the primary reasons for this include the difficulty in maintaining accurate pronunciation while preserving the same duration before and after translation, as well as the limited training data for Fr and De. Achieving more robust cross-lingual translation remains a focus for future work. We also show some examples of speech translation in our demo page.

---

[11]`https://huggingface.co/pyp1/VoiceCraft/blob/main/830M_TTSEnhanced.pth`
[12]`https://huggingface.co/model-scope/CosyVoice-300M`
[13]`ttps://huggingface.co/openai/whisper-medium`

Table 15: Evaluation results in cross-lingual speech translation with consistent total duration.

| | Zh | | En | | Ja | | Ko | | De | | Fr | |
|---|---|---|---|---|---|---|---|---|---|---|---|---|
| | WER | SIM-O | WER | SIM-O | WER | SIM-O | WER | SIM-O | WER | SIM-O | WER | SIM-O |
| Zh | - | - | 7.466 | 0.678 | 7.864 | 0.720 | 9.751 | 0.736 | 25.54 | 0.724 | 16.21 | 0.687 |
| En | 7.411 | 0.535 | - | - | 5.870 | 0.544 | 12.18 | 0.543 | 12.43 | 0.579 | 17.48 | 0.590 |
| Ja | 13.93 | 0.647 | 7.387 | 0.642 | - | - | 10.98 | 0.703 | 12.85 | 0.649 | 14.61 | 0.645 |
| Ko | 31.30 | 0.734 | 14.61 | 0.697 | 12.79 | 0.749 | - | - | 26.58 | 0.722 | 33.96 | 0.712 |
| De | 19.54 | 0.714 | 5.148 | 0.740 | 6.072 | 0.678 | 12.02 | 0.667 | - | - | 14.53 | 0.672 |
| Fr | 32.84 | 0.672 | 12.17 | 0.682 | 6.076 | 0.640 | 12.07 | 0.582 | 21.65 | 0.682 | - | - |

## G  POST-TRAINING FOR EMOTION CONTROL

MaskGCT can unlock more extensive capabilities with post-training. We take emotion control as an example. After being pretrained on a large-scale dataset, we fine-tune the T2S model by adding an additional emotion label as a prefix to the original input sequence. We use an emotion dataset, ESD [61], which consists of 350 parallel utterances with an average duration of 2.9 seconds spoken by 10 native English and 10 native Mandarin speakers, to fine-tune our model. The experimental results show that MaskGCT can unlock emotion control capabilities for zero-shot in-context learning scenarios. For the construction of the train and test datasets, we selected one male and one female speaker each from native English and native Mandarin backgrounds, resulting in a total of four speakers for the test dataset. The remaining 16 speakers were allocated to the training dataset. For the 350 parallel Chinese utterances, we randomly chose 22 utterances for the test set, with the remaining utterances designated for training. Similarly, for the 350 parallel English utterances, we randomly selected 21 utterances for the test set, with the rest used for training. To assess the consistency between the generated audio and the target emotion label, we trained an emotion classification model using the constructed train dataset. This model achieved a classification accuracy of 72% on the test dataset. We show some examples in our demo page.

## H  SPEECH CONTENT EDITING

Based on the mask-and-predict mechanism, our text-to-semantic model supports zero-shot speech content editing with the assistance of a text-speech aligner. By using the aligner, we can identify the editing boundary of the original semantic token sequence, mask the portion that needs to be edited, and then predict the masked semantic tokens using the edited text and the unmasked semantic tokens. However, we have observed that our system is not very robust in editing tasks. A possible conjecture is that we need to adopt a training paradigm better suited for editing tasks, such as fill-in-mask [9, 74]. We show some examples in our demo page.

## I  VOICE CONVERSION

MaskGCT supports zero-shot voice conversion by fine-tuning the S2A with a modified training strategy. The zero-shot voice conversion task aims to alter the source speech to sound like that of a target speaker using a reference speech from the target speaker, without changing the semantic content. We can directly use the semantic tokens $\mathbf{S}_{src}$ extracted from the source speech and the prompt acoustic tokens $\mathbf{A}_{ref}$ extracted from the reference speech to predict the target acoustic tokens $\mathbf{A}_{tgt}$. Since $\mathbf{S}_{src}$ may retain some timbre information, we perform timbral perturbation on the semantic features input to the semantic codec encoder. Specifically, we apply timbral perturbation to the input mel-spectrogram features of the W2v-BERT 2.0 model, following the method outlined in FreeVC [75]. We fine-tune our S2A model using this training strategy. We show some examples in our demo page.

## J  HARD CASES EVALUATION

We evaluate the performance of MaskGCT on some hard cases (SeedTTS *test-hard*), which refer to instances where large-scale TTS models, particularly those AR-based models, often exhibit hallucinations. These cases include phrases with repeating words, tongue twisters, and other complex

linguistic structures. Examples of such cases include: "*the great greek grape growers grow great greek grapes*", "*How many cookies could a good cook cook If a good cook could cook cookies? A good cook could cook as much cookies as a good cook who could cook cookies*", and " *thought a thought. But the thought I thought wasn't the thought I thought I thought. If the thought I thought I thought had been the thought I thought, I wouldn't have thought so much*".

Table 16: The evaluation results of MaskGCT and AR + SoundStorm on SeedTTS *test-hard*.

| System | SIM-O ↑ | WER ↓ |
|---|---|---|
| **SeedTTS *test-hard*** | | |
| AR + SoundStorm | 0.692 | 34.16 |
| AR + SoundStorm (*rank 5*) | 0.739 | 17.05 |
| MaskGCT | 0.748 | 10.27 |
| MaskGCT (*rank 5*) | **0.776** | **6.258** |

## K  DISCUSSION ABOUT CONCURRENT WORKS

SimpleSpeech [30], DiTTo-TTS [31], E2 TTS [32], and F5-TTS [76] are also NAR-based models that do not necessitate precise alignment information between text and speech, nor do they forecast phoneme-level duration. These are concurrent works with MaskGCT. The three models all employ diffusion modeling on speech representations within continuous spaces. SimpleSpeech models the latent representation of a wav codec based on finite scalar quantization (FSQ) [77], DiTTo-TTS utilizes the latent representation of a wav codec based on residual vector quantization (RVQ), and E2 TTS and F5-TTS directly model the mel-spectrogram with flow matching. However, Both F5-TTS and A2Flow [78] mention that direct modeling mel-spectrogram is characterized by slow convergence and difficulty in achieving convergence on small datasets. To further investigate this issue, we train the T2S models on small datasets (LibriTTS [79] and a 1K hours subset of Emilia) while reusing the S2A model (which is entirely self-supervised and does not require text). The results in Table 17 demonstrate that our model performs well even when trained on small datasets. This is likely due to the ease of predicting semantic tokens and the powerful modeling capabilities of masked generative models.

Table 17: The evaluation results of MaskGCT on small training datasets.

| Model | SIM-O ↑ | WER ↓ |
|---|---|---|
| **SeedTTS *test-en*** | | |
| MaskGCT (LibriTTS 0.58K hours) | 0.677 | 3.043 |
| MaskGCT (Emilia 1K hours) | 0.696 | 3.378 |
| MaskGCT (Emilia 100K hours) | 0.728 | 2.466 |
| **SeedTTS *test-zh*** | | |
| MaskGCT (Emilia 1K hours) | 0.754 | 3.012 |
| MaskGCT (Emilia 100K hours) | 0.777 | 2.183 |

We also compare MaskGCT with a direct text-to-acoustic approach using masked generative models (which can be seen as removing semantic tokens as a condition and adding text as a condition based on the S2A model) on a 10K hours subset. The results in Table 18 indicate that directly predicting acoustic tokens from text is challenging to converge, resulting in lower SIM and significantly higher WER, demonstrating that the two-stage model reduces the overall modeling difficulty.

Table 18: Comparison of MaskGCT and Text-to-Acoustic.

| Model | SIM-O ↑ | WER ↓ |
|---|---|---|
| **SeedTTS *test-en*** | | |
| Text-to-Acoustic (Emilia 10K hours) | 0.651 | 12.75 |
| MaskGCT (Emilia 10K hours) | **0.719** | **2.872** |
| **SeedTTS *test-zh*** | | |
| Text-to-Acoustic (Emilia 10K hours) | 0.727 | 17.08 |
| MaskGCT (Emilia 10K hours) | **0.762** | **3.302** |

## L    BOARDER IMPACT

Given that our model can synthesize speech with high speaker similarity, it carries potential risks of misuse, including spoofing voice identification or impersonating specific speakers. Our experiments were conducted under the assumption that the user consents to be the target speaker for speech synthesis. To mitigate misuse, it is essential to develop a robust model for detecting synthesized speech and to establish a system for reporting suspected misuse.

