# OpenReview forum: "MaskGCT: Zero-Shot Text-to-Speech with Masked Generative Codec Transformer"
_ICLR.cc/2025/Conference — ICLR 2025 Poster_

### Official Review · Reviewer_EzBH · 2024-10-27

**Soundness:** 2
**Presentation:** 3
**Contribution:** 2
**Rating:** 6
**Confidence:** 4

**Summary:**

This paper presents MASKGCT, a compact Non-Autoregressive (NAR) text-to-speech model. This model is meticulously designed with a specific emphasis on attaining outstanding audio quality. What sets it apart from prior works is its utilization of semantic tokens extracted from the SSL representation using VQVAE and its avoidance of relying on explicit alignment information.

**Strengths:**

- This paper offers quite interesting and remarkably high-quality demo audio. The audio samples provided within the paper not only demonstrate clear and distinct sound qualities but also exhibit a certain level of innovation in terms of the audio characteristics they present. They are able to effectively capture the attention of the readers and give a practical sense of the research outcomes in the audio domain.
- The proposed pipeline is significantly more compact than previous ones such as NaturalSpeech3. This compactness implies a more streamlined and efficient design. It potentially leads to reduced computational complexity and may offer advantages in terms of implementation and resource utilization.

**Weaknesses:**

- The Word Error Rate (WER) in Table 4 and 5 of all methods is much higher than previously reported. This discrepancy requires more in-depth explanation. It is essential to analyze the factors contributing to this higher WER, such as possible differences in the data sets used, the experimental settings, or any unique characteristics of the methods employed in this study. A detailed exploration and clarification of these aspects would enhance the understanding and validity of the results presented.
- In Section 3.2.1, the authors utilize VQVAE to obtain discrete semantic tokens instead of k-means. However, no corresponding experiment result is provided. It would be beneficial to include the experimental results related to the use of k-means for a more comprehensive understanding. These results could demonstrate the effectiveness and performance of VQVAE in this context, and also facilitate a comparison with the potential outcomes if k-means had been used. This would add more substance to the discussion and support the authors' choice of using VQVAE.
- To my knowledge, using text as a condition without expanding by duration for a Non-Autoregressive (NAR) model was first proposed in e2-tts. The authors should point out this in Section 3.2.2. Acknowledging the prior work in this area is important for providing a complete context and showing the evolution of the research. By highlighting the connection to e2-tts, the authors can better position their work within the existing body of knowledge and give credit to the relevant pioneering research. It also helps readers to better understand the background and significance of the approach taken in this study.

**Questions:**

Address my concerns in the Weaknesses.

---

> ### Author Response · Authors · 2024-11-17
> **Reply to Reviewer EzBH (Part 1)**
>
> First of all, we want to thank the reviewer for your careful reading and providing a lot of constructive comments! Below we address the concerns mentioned in the review.
>
> ## Weaknesses
>
> ``Weakness 1: The higher WER values in Tables 4 and 5 compared to previous reports need explanation.``
>
> We find that one possible reason for the high WER could be that the ASR model has poor recognition capabilities for accents and emotional data. We discovered that the WER of the ground truth is also high. Thanks for your valuable suggestions, we will incorporate these discussions into the revised version of our paper.
>
> ``Weakness 2: The choice of VQ-VAE over k-means for semantic tokenization needs experimental validation.``
>
> Thank you for your valuable suggestion. This question has also been raised by other reviewers, and we believe a thorough investigation of this matter will significantly strengthen our paper. We will first present our empirical findings (which are already included in our paper), followed by additional experimental results. **All these analyses will be incorporated into the revised version of our paper.**
>
> In our initial experiments, we observed that k-means-based semantic tokens were less effective in predicting acoustic tokens for languages with rich prosodic variations, particularly Chinese, where significant pitch variations were observed. We further support this finding with qualitative analysis.
>
> 1. Since k-means can be seen as optimizing the reconstruction loss between input and reconstruction, we present reconstruction loss curves under different k-means and VQ configurations.
> We compare four configurations: VQ-8192 (which is the same as in our paper), VQ-2048, k-means-8192, and k-means-2048.
> The loss curves can be found at this [link](https://github.com/maskgct/maskgct/raw/refs/heads/main/recon_loss_for_vq.PNG).
>
> 2. The information preservation in semantic tokens directly affects the semantic-to-acoustic model's prediction performance. We present the top-10 accuracy (shown in this [link](https://raw.githubusercontent.com/maskgct/maskgct/refs/heads/main/soundstorm_layer1_acc.PNG)) of the semantic-to-acoustic model in predicting the first layer of acoustic tokens. The results demonstrate that VQ-8192 outperforms VQ-2048, which in turn outperforms k-means-8192.
>
> 3. We investigate the impact of different semantic representation approaches on acoustic token reconstruction. We train separate semantic-to-acoustic (S2A) models for each configuration and evaluate their performance through speech reconstruction metrics. Across all three test sets, the results reveal a consistent performance ranking in similarity (SIM) scores, with VQ-8192 yielding the highest performance, followed sequentially by VQ-2048, k-means-8192, and k-means-2048. For WER, VQ-based methods also demonstrate superior performance over k-means approaches, though this advantage is less pronounced on LibriSpeech test-clean. Notably, on SeedTTS test-zh (Chinese test set), k-means exhibits a substantial degradation in WER. We attribute this to the stronger coupling between Chinese semantics and prosodic features, where the transition from VQ to k-means results in significant loss of prosodic information in the semantic representations.
>
> | Semantic Codec | LibriSpeech *test-clean* | | SeedTTS *test-en* | | SeedTTS *test-zh* | |
> |--------------|----------|---------|----------|---------|----------|---------|
> | | SIM-O ↑ | WER ↓ | SIM-O ↑ | WER ↓ | SIM-O ↑ | WER ↓ |
> | k-means 2048 | 0.648 | 3.013 | 0.658 | 3.989 | 0.691 | 11.420 |
> | k-means 8192 | 0.661 | 2.862 | 0.664 | 3.012 | 0.713 | 8.782 |
> | VQ 2048 | *0.671* | **2.177** | *0.692* | *3.187* | *0.744* | *4.913* |
> | VQ 8192 | **0.680** | *2.223* | **0.713** | **2.175** | **0.763** | **2.088** |
>
>
> These comprehensive analyses demonstrate the advantages of VQ-based semantic tokens over k-means clustering, providing strong empirical evidence to support our design choices in MaskGCT.

---

> ### Author Response · Authors · 2024-11-17
> **Reply to Reviewer EzBH (Part 2)**
>
> ``Weakness 3: should acknowledge e2-tts as the first work using unexpanded text conditioning in NAR TTS.``
>
> Thank you for highlighting this important point. We would like to clarify our treatment of this topic:
>
> Thank you for pointing out a very important point. In our related works, we mention that "SimpleSpeech, DiTTo-TTS, and E2-TTS [1] are also NAR-based models that do not require precise alignment information between text and speech, nor do they predict phone-level duration," and discuss the differences between our model and theirs in Appendix K. E2-TTS and its recent open-source version [2] have demonstrated superior performance in large-scale training. However, recent work [2, 3] also points out that it struggles to converge on small datasets. We conduct a similar experiment, directly predicting acoustic codes from text, and observed similar phenomena. We provide some results below. The results show that directly predicting acoustic codes from text is challenging to converge, resulting in lower SIM and significantly higher WER. Of course, this approach differs from E2-TTS in several aspects. We conduct this investigation primarily to demonstrate that our two-stage model potentially reduces the overall modeling complexity.
>
> | Model | SIM-O ↑ | WER ↓ |
> |-------|----------|---------|
> ||**SeedTTS *test-en***|
> | Text-to-Acoustic (Emilia 10K hours) | 0.651 | 12.75 |
> | MaskGCT (Emilia 10K hours) | **0.719** | **2.872** |
> || **SeedTTS *test-zh*** |
> | Text-to-Acoustic (Emilia 10K hours) | 0.727 | 17.08 |
> | MaskGCT (Emilia 10K hours) | **0.762** | **3.302** |
>
> Thanks again for your constructive comments. We would be grateful if we could hear your feedback regarding our answers to the reviews. We would be happy to answer and discuss if you have further comments.
>
> [1] Eskimez S E, Wang X, Thakker M, et al. E2 TTS: Embarrassingly easy fully non-autoregressive zero-shot TTS[J]. arXiv preprint arXiv:2406.18009, 2024.
>
> [2] Chen Y, Niu Z, Ma Z, et al. F5-TTS: A Fairytaler that Fakes Fluent and Faithful Speech with Flow Matching[J]. arXiv preprint arXiv:2410.06885, 2024.
>
> [3] A^2-Flow: Alignment-Aware Pre-training for Speech Synthesis with Flow Matching https://openreview.net/attachment?id=e2p1BWR3vq&name=pdf

---

> > ### Author Response · Authors · 2024-11-20
> >
> > Thanks again for your valuable comments, we would be grateful if we could hear your feedback regarding our answers. We would be happy to answer and discuss if you have further comments.

---

> > > ### Author Response · Authors · 2024-11-22
> > >
> > > Thank you again for your great efforts and the valuable comments.
> > >
> > > We have carefully addressed the main concerns in detail. We hope you might find the response satisfactory. As the paper discussion phase is coming to an end, we would be grateful if we could hear your feedback regarding our answers to the reviews. We will be very happy to clarify any remaining points (if any).

---

> > > > ### Comment · Reviewer_EzBH · 2024-11-27
> > > >
> > > > Thank you for your response. While Reviewer n456's comments on novelty might be overly critical, the authors have provided excellent video demonstrations, particularly the one featuring Black Myth. However, considering the overall quality of the paper, I believe maintaining my current score is appropriate.

---

> > > > > ### Author Response · Authors · 2024-11-27
> > > > >
> > > > > Thank you for your feedback! If you have any remaining concerns or suggestions to help further enhance this research, please feel free to comment, and we are more than willing to continue addressing these issues.

---

### Official Review · Reviewer_AvMh · 2024-10-27

**Soundness:** 3
**Presentation:** 3
**Contribution:** 3
**Rating:** 6
**Confidence:** 4

**Summary:**

The authors present a zero-short text-to-speech (TTS) system called Masked Generative Codec Transformer (MaskGCT), which comprises two stages: 1) predicting semantic tokens from text and 2) using these semantic tokens to generate acoustic tokens. Both stages leverage a non-autoregressive masked generative transformer with specific prompt tokens for in-context learning. A VQ-VAE codec and a residual vector quantization (RVQ) codec are separately trained to obtain the semantic tokens and acoustic tokens. Extensive experiments across various tasks demonstrate that MaskGCT, trained on 100k hours of speech data, surpasses existing state-of-the-art zero-shot TTS systems in quality, similarity, and intelligibility.

**Strengths:**

- This work utilizes non-autoregressive masked generative transformers to generate both sematic tokens and acoustic tokens, offering advantages such as flexible length control, better robustness, and higher inference efficiency. This is suggested by the experiments in Section 4.2.2, comparing to the system replacing the first stage with an autoregressive model (i.e., AR + SoundStorm).
- The authors conduct extensive experiments on a variety of tasks to evaluate the effectiveness and sca of the proposed system, including not only standard zero-shot TTS, but also cross-lingual dubbing, voice conversion, emotion control, and speech content editing, with potential use of post-training and fine-tuning.
- The selected state-of-the-art baselines are well-chosen.

**Weaknesses:**

- The novelty of this work is somewhat incremental, though I would appreciate its engineering contributions. This work follows a two-stage codec-based sequence-to-sequence strategy, converting text into semantic tokens and then into acoustic tokens, which is in line with SPEAR-TTS [1]. The SoundStorm [2] paper also explores this strategy. Additionally, the idea of applying non-autoregressive masked generative transformer for codec-based TTS is presented in SoundStorm [2] and NaturalSpeech3 [3] (though the NaturalSpeech3 paper describes it from a diffusion perspective.). However, I would like to highlight the distinction that this work employs non-autoregressive masked generative transformers in both stages of the two-stage framework.
- The authors emphasize that the proposed method does not require text-speech alignment supervision or phone-level duration prediction. However, it appears that the method does require a specified length for the target semantic token sequence, thereby to control the length of the generate speech. The presented experimental results in Table 2, comparing to baseline methods, are based on either ground-truth duration or predicted duration. As descibed in the paper, the authors train a duration predictor to estimate the phone-level durations, which are then summed to determine the total duration of the target speech. This process relies on pre-computed ground-truth phone-level durations. Therefore, it seems that this may not fully align with the claim of not requiring text-speech alignment supervision or phone-level duration prediction.
- The authors claim that training a VQ-VAE model to quantize the semantic representation can reduce the information loss, compared to using k-means clustering as in previous works. I think it would be better if the authors can provide ablation results or relevant references to support this assertion.

[1] Kharitonov, E., Vincent, D., Borsos, Z., Marinier, R., Girgin, S., Pietquin, O., ... & Zeghidour, N. (2023). Speak, read and prompt: High-fidelity text-to-speech with minimal supervision. Transactions of the Association for Computational Linguistics, 11, 1703-1718.

[2] Borsos, Z., Sharifi, M., Vincent, D., Kharitonov, E., Zeghidour, N., & Tagliasacchi, M. (2023). Soundstorm: Efficient parallel audio generation. arXiv preprint arXiv:2305.09636.

[3] Ju, Z., Wang, Y., Shen, K., Tan, X., Xin, D., Yang, D., ... & Zhao, S. (2024). Naturalspeech 3: Zero-shot speech synthesis with factorized codec and diffusion models. arXiv preprint arXiv:2403.03100.

**Questions:**

- In Table 1, does "ZS TTS" denote "zero-shot TTS", and what's "CL TTS"? For "Imp. Dur.", I also don't fully agree that this work implicitly models duration, as it requires a specified length of the target generated speech.
- In Section 3.2.3,  why "the number of frames in the semantic token sequence is equal to the sum of the frames in the prompt acoustic sequence and the target acoustic sequence"?
- Is the output length of the text-to-semantic stage equal to that from the semantic-to-acoustic stage? From Figure 2, the former seems shorter than the latter?
- In the first paragraph of Section 2, about "SpearTTS utilizes three AR models to predict semantic tokens from text, coarse-grained acoustic tokens from semantic tokens, and fine-grained acoustic tokens from coarse-grained tokens.", the terms "coarse-grained" and "fine-grained" do not seem to be mentioned in the SpearTTS [1] paper, what do the authors mean by them?
- What's the length of prompt audio? Does it affect the quality of the generated speech?

[1] Kharitonov, E., Vincent, D., Borsos, Z., Marinier, R., Girgin, S., Pietquin, O., ... & Zeghidour, N. (2023). Speak, read and prompt: High-fidelity text-to-speech with minimal supervision. Transactions of the Association for Computational Linguistics, 11, 1703-1718.

---

> ### Author Response · Authors · 2024-11-17
> **Reply to Reviewer AvMh (Part 1)**
>
> First of all, we want to thank the reviewer for your careful reading and providing a lot of constructive comments! Additionally, we appreciate the acknowledgment of the impact of our work on the community. Below we address the concerns mentioned in the review.
>
> ## Weaknesses
> ``Weakness 1: The novelty is incremental as it mainly follows SPEAR-TTS's two-stage strategy and SoundStorm/NaturalSpeech3's masked generation approach, though it uniquely applies non-AR masked generation to both stages.``
>
> We appreciate your recognition of our engineering contributions and **commit to open-sourcing all code and weights to benefit the research community**. We would like to highlight MaskGCT's significant contributions at both representation and modeling levels:
>
> 1. At the modeling level, MaskGCT innovatively applies masked generative modeling to both text-to-semantic and semantic-to-acoustic generation. While this approach was originally developed for image generation (MaskGIT), its adaptation to TTS presents unique challenges and opportunities. Notably, MaskGCT achieves non-autoregressive TTS without requiring explicit phone-speech alignment supervision or phone-level duration prediction, which significantly simplifies the traditional TTS pipeline while achieving more natural and consistent generation results.
>
> 2. At the representation level, MaskGCT introduces VQ-based semantic tokens, which represents a distinct approach compared to previous methods that relied on k-means clustering.
>
> ``Weakness 2: The claim of not requiring text-speech alignment or duration prediction seems inconsistent, as the method still needs target sequence length prediction based on phone-level durations.``
>
> It is noteworthy that there are several methods to determine the total duration length. Our trained duration predictor is solely for providing a rough estimate to facilitate inference and comparison. Alternatively, a simpler approach could be: $\textit{target total duration} = \textit{target phone number} \times \frac{\textit{prompt total duration}}{\textit{prompt phone number}}$. Additionally, one could train a regression-based model to directly predict the total duration using prompt text, target text, and prompt speech as inputs, as proposed in [1]. In Section 4.2.3 and our demo pages, we also demonstrate that our method can generate satisfactory results within a reasonable range of total durations.
>
> The table below illustrates the comparative results of MaskGCT under three different total duration calculation methods. The results indicate that our model, using simple rules to predict total duration, can generate speech with SIM and WER that are essentially comparable to those of the ground truth.
>
> | Method | SIM-O ↑ | WER ↓ |
> |--------|----------|---------|
> || **LibriSpeech *test-clean*** ||
> | *rule-based* | 0.686 | 2.976 |
> | *duration predictor* | 0.687 | 2.634 |
> | *gt length* | 0.697 | 2.012 |
> || **SeedTTS *test-en*** ||
> | *rule-based* | 0.719 | 2.712 |
> | *duration predictor* | 0.717 | 2.623 |
> | *gt length* | 0.728 | 2.466 |
> || **SeedTTS *test-zh*** ||
> | *rule-based* | 0.771 | 2.409 |
> | *duration predictor* | 0.774 | 2.273 |
> | *gt length* | 0.777 | 2.183 |

---

> > ### Comment · Reviewer_AvMh · 2024-11-25
> > **Thanks for the detailed response**
> >
> > Thanks for the detailed reply!
> >
> > I think the supplemented experiments comparing different methods for quantizing the semantic representation ( VQ-VAE vs. k-means clustering) have addressed my concern.
> >
> > However, I still have two main concerns:
> >
> > - There are already codec-based TTS works, such as SoundStorm and NaturalSpeech3, that apply non-autoregressive masked generative modeling. Therefore, I still have concerns regarding the novelty of this aspect.
> > - The authors have supplemented experiments using a rule-based method to determine the total duration length for inference, which I agree does not require text-speech alignment supervision or phone-level duration prediction in training. However, the results in Table 1 comparing the proposed system with other state-of-the-art models do not present this method. I think this is important, as the authors emphasize a key contribution of this work in abstract: "eliminates the need for explicit alignment information between text and speech supervision, as well as phone-level duration prediction".
> >
> > Please let me know if I have misunderstood.

---

> ### Author Response · Authors · 2024-11-17
> **Reply to Reviewer AvMh (Part 2)**
>
> ``Weakness 3: The claim about VQ-VAE's superiority over k-means clustering for semantic representation needs experimental validation.``
>
> Thank you for your valuable suggestion. This question has also been raised by other reviewers, and we believe a thorough investigation of this matter will significantly strengthen our paper. We will first present our empirical findings (which are already included in our paper), followed by additional experimental results. **All these analyses will be incorporated into the revised version of our paper.**
>
> In our initial experiments, we observed that k-means-based semantic tokens were less effective in predicting acoustic tokens for languages with rich prosodic variations, particularly Chinese, where significant pitch variations were observed. We further support this finding with qualitative analysis.
>
> 1. Since k-means can be seen as optimizing the reconstruction loss between input and reconstruction, we present reconstruction loss curves under different k-means and VQ configurations.
> We compare four configurations: VQ-8192 (which is the same as in our paper), VQ-2048, k-means-8192, and k-means-2048.
> The loss curves can be found at this [link](https://github.com/maskgct/maskgct/raw/refs/heads/main/recon_loss_for_vq.PNG).
>
> 2. The information preservation in semantic tokens directly affects the semantic-to-acoustic model's prediction performance. We present the top-10 accuracy (shown in this [link](https://raw.githubusercontent.com/maskgct/maskgct/refs/heads/main/soundstorm_layer1_acc.PNG)) of the semantic-to-acoustic model in predicting the first layer of acoustic tokens. The results demonstrate that VQ-8192 outperforms VQ-2048, which in turn outperforms k-means-8192.
>
> 3. We investigate the impact of different semantic representation approaches on acoustic token reconstruction. We train separate semantic-to-acoustic (S2A) models for each configuration and evaluate their performance through speech reconstruction metrics. Across all three test sets, the results reveal a consistent performance ranking in similarity (SIM) scores, with VQ-8192 yielding the highest performance, followed sequentially by VQ-2048, k-means-8192, and k-means-2048. For WER, VQ-based methods also demonstrate superior performance over k-means approaches, though this advantage is less pronounced on LibriSpeech test-clean. Notably, on SeedTTS test-zh (Chinese test set), k-means exhibits a substantial degradation in WER. We attribute this to the stronger coupling between Chinese semantics and prosodic features, where the transition from VQ to k-means results in significant loss of prosodic information in the semantic representations.
>
> | Semantic Codec | LibriSpeech *test-clean* | | SeedTTS *test-en* | | SeedTTS *test-zh* | |
> |--------------|----------|---------|----------|---------|----------|---------|
> | | SIM-O ↑ | WER ↓ | SIM-O ↑ | WER ↓ | SIM-O ↑ | WER ↓ |
> | k-means 2048 | 0.648 | 3.013 | 0.658 | 3.989 | 0.691 | 11.420 |
> | k-means 8192 | 0.661 | 2.862 | 0.664 | 3.012 | 0.713 | 8.782 |
> | VQ 2048 | *0.671* | **2.177** | *0.692* | *3.187* | *0.744* | *4.913* |
> | VQ 8192 | **0.680** | *2.223* | **0.713** | **2.175** | **0.763** | **2.088** |
>
> We believe these additions will strengthen our argument and offer a clearer understanding of the impact of different semantic token approaches on model performance.
>
> ## Questions
>
> ``Q1: In Table 1, does "ZS TTS" denote "zero-shot TTS", and what's "CL TTS"? For "Imp. Dur.", I also don't fully agree that this work implicitly models duration, as it requires a specified length of the target generated speech.``
>
> Yes, "ZS TTS" denotes "zero-shot", "CL TTS" denotes cross-lingual TTS. For the latter part of the question, we are primarily referring to the absence of explicit phone-level duration modeling. We appreciate the feedback on the unclear definitions. We will add more detailed explanations in the revised paper.
>
> ``Q2: In Section 3.2.3, why "the number of frames in the semantic token sequence is equal to the sum of the frames in the prompt acoustic sequence and the target acoustic sequence"?``
>
> Because our semantic tokens and acoustic tokens both have a frame rate of 50Hz, while the acoustic tokens consist of multiple layers, each with a frame rate of 50Hz. More accurately, the semantic token sequence input to the semantic-to-acoustic model is a concatenation of the semantic tokens corresponding to the prompt speech and the target speech.
>
> It's worth noting that semantic and acoustic tokens don't necessarily need to share the same frame rate, as long as they maintain a fixed ratio of tokens generated per unit time.

---

> ### Author Response · Authors · 2024-11-17
> **Reply to Reviewer AvMh (Part 3)**
>
> ``Q3: Is the output length of the text-to-semantic stage equal to that from the semantic-to-acoustic stage? From Figure 2, the former seems shorter than the latter?``
>
> Yes, they are the same. More precisely, the total length of the prompt semantic tokens and target semantic tokens is consistent with the total length of the prompt acoustic tokens and target acoustic tokens.
>
> ``Q4: The terms "coarse-grained" and "fine-grained" do not seem to be mentioned in the SpearTTS paper, what do the authors mean by them?``
>
> In our paper, "coarse-grained" refers to the first layer of acoustic tokens, while "fine-grained" refers to the remaining layers. This terminology aligns with AudioLM [2], which SpearTTS extends. AudioLM explicitly uses the terms "Coarse acoustic modeling" and "Fine acoustic modeling" in its framework.
> To clarify the connection: SpearTTS employs an autoregressive (AR) model to predict semantic codes from text, followed by another AR model for acoustic code prediction. As stated in the SpearTTS paper: "Similar to AudioLM, it is possible to add an optional third stage, with the goal of improving the quality of the synthesized speech by predicting acoustic tokens corresponding to fine residual vector quantization levels."
>
> ``Q5: What's the length of prompt audio? Does it affect the quality of the generated speech?``
>
> Our model offers considerable flexibility in prompt handling:
>
> 1. Training Strategy: We randomly select 0-50% of the input speech as the prompt during training, enabling the model to adapt to varying prompt lengths.
>
> 2. Inference Guidelines: Since our training data consists of samples under 30 seconds, we recommend keeping prompt lengths below 20 seconds for optimal performance in practical applications.
>
> Thanks again for your constructive comments. We would be grateful if we could hear your feedback regarding our answers to the reviews. We would be happy to answer and discuss if you have further comments.
>
> [1] Lee K, Kim D W, Kim J, et al. DiTTo-TTS: Efficient and Scalable Zero-Shot Text-to-Speech with Diffusion Transformer[J]. arXiv preprint arXiv:2406.11427, 2024.
>
> [2] Borsos Z, Marinier R, Vincent D, et al. Audiolm: a language modeling approach to audio generation[J]. IEEE/ACM transactions on audio, speech, and language processing, 2023, 31: 2523-2533.

---

> > ### Author Response · Authors · 2024-11-20
> >
> > Thanks again for your valuable comments, we would be grateful if we could hear your feedback regarding our answers. We would be happy to answer and discuss if you have further comments.

---

> > > ### Author Response · Authors · 2024-11-22
> > >
> > > Thank you again for your great efforts and the valuable comments.
> > >
> > > We have carefully addressed the main concerns in detail. We hope you might find the response satisfactory. As the paper discussion phase is coming to an end, we would be grateful if we could hear your feedback regarding our answers to the reviews. We will be very happy to clarify any remaining points (if any).

---

> ### Author Response · Authors · 2024-11-27
>
> Thank you very much for your valuable feedback!
>
> - I agree that both SoundStorm and NaturalSpeech 3 are codec-based TTS systems. In SoundStorm, it uses an AR (Autoregressive) model as the first stage and an NAR (Non-Autoregressive) model as the second stage. Robustness and inference speed are known issues in AR models. Unlike AR-based models, NaturalSpeech 3 explicitly employs a phone duration predictor and then obtains the frame-level phoneme as a condition to predict the frame-level prosody tokens, followed by predicting the frame-level content tokens and acoustic tokens. This means it requires pre-extracted phone duration for model training, and the performance of phone duration prediction impacts the overall prosody of the generated speech. Unlike SoundStorm and NaturalSpeech 3, MaskGCT does not use phone-level duration. We only need phone-level conditioning to predict frame-level semantic tokens, or directly use text tokens obtained from BPE (we have shown the results in Appendix A.7, which indicate that using BPE does not cause significant performance loss and may even yield better WER for Chinese). Additionally, MaskGCT also explores speech semantic tokens based on VQ. In our supplementary experiments, we also investigate the differences between a two-stage model and direct text-to-acoustic modeling (the results are shown in Appendix K).
>
> - We also illustrate the differences between MaskGCT and traditional NAR (Non-Autoregressive) methods in text-speech alignment modeling through a figure (in this [https://raw.githubusercontent.com/maskgct/maskgct/refs/heads/main/aatn_map.png](https://raw.githubusercontent.com/maskgct/maskgct/refs/heads/main/aatn_map.png)). For traditional NAR TTS systems, a duration predictor and length regulator are first used to obtain frame-level conditions that provide alignment information. For MaskGCT, similar to AR (Autoregressive) methods, we concatenate the text in front of the speech tokens through in-context learning, and the model implicitly learns speech-text alignment through self-attention.
>
> - In addition, we provide attention maps at different stages of inference (steps 1, 10, 20) and across various layers of the model (layer 1, layer 6, layer 16) in this [https://raw.githubusercontent.com/maskgct/maskgct/refs/heads/main/ar_vs_nar_vs_maskgct.png](https://raw.githubusercontent.com/maskgct/maskgct/refs/heads/main/ar_vs_nar_vs_maskgct.png). These attention maps demonstrate that our model implicitly learns the alignment between text and speech. It shows that the model handles the alignment between speech tokens and text in the middle layers of the network, while in the deeper layers, the model has already resolved the alignment and focuses on processing the tokens.
>
> - For the second concern, we appreciate the highly constructive comments, and we will incorporate the results with the rule-based duration into Table 1, as we agree that this can better substantiate our claim that text-speech alignment supervision or phone-level duration prediction is unnecessary. We further emphasize and discuss this point in Appendix A.5. Additionally, the absence of phone-level duration means that we can simply adjust the total duration to control the tempo of the generated speech. In Section 4.2.3 Duration Length Analysis, we demonstrate that MaskGCT exhibits robustness across gt duration lengths ranging from gt len * 0.7 to gt len * 1.3, and at the demo page "Speech Tempo Controllability", we showcase the demonstrate of the produced speech across varying total durations (gt len * 0.6 to gt len * 1.2), all of which exhibit good naturalness.
>
> Thanks again for your valuable comments. We have made every effort to address your concerns and enhance our paper, and we kindly hope you can reconsider our paper. If you have any further suggestions or concerns, we are more than willing to provide more discussion.

---

> > ### Author Response · Authors · 2024-11-29
> >
> > Dear Reviewer, Once again, we would like to express our sincere gratitude for your valuable and constructive feedback. We would like to inquire whether our responses have further addressed your concerns. As the rebuttal phase is nearing its end, we would appreciate it if you could let us know if there are any further concerns or suggestions for improving our paper. We are more than willing to address any issues promptly. Thank you very much.

---

> > > ### Author Response · Authors · 2024-12-03
> > >
> > > Thanks again for your valuable and constructive feedback. Since the rebuttal phase is nearing its end, we would appreciate it if you could let us know whether our further reply has addressed your concern. If you have any further suggestions or concerns, we are more than willing to provide more discussion as soon as possible.

---

> > > > ### Comment · Reviewer_AvMh · 2024-12-03
> > > > **Thanks for response**
> > > >
> > > > I believe this paper is overall well-completed, and I will maintain my score at 6.

---

### Official Review · Reviewer_n456 · 2024-11-02

**Soundness:** 3
**Presentation:** 3
**Contribution:** 2
**Rating:** 3
**Confidence:** 5

**Summary:**

This paper introduces MaskGCT, a non-autoregressive (NAR) masked generative modeling method that sequentially generates audio from acoustic tokens, semantic tokens, and text. The semantic tokens are derived from VQ-VAE of W2V-BERT, a self-supervised speech learning model. The acoustic token approach extends DAC with improved training efficiency.

Results demonstrate that MaskGCT either outperforms or matches current autoregressive (AR) based methods and NAR duration-based methods with publicly available implementation details, across multiple benchmarks: LibriSpeech, SeedTTS (Chinese and English), zero-shot TTS, and speech style (timbre, emotion, accent, etc.) imitation learning. Additionally, MaskGCT surpasses AR-based methods in both naturalness and computational efficiency (inference time steps).

**Strengths:**

* This paper experimentally demonstrates that NAR (non-autoregressive) non-duration-based TTS methods outperform both AR (autoregressive) based and NAR duration-based TTS methods with publicly available implementation details.

* The open-source implementation and comprehensive reproducibility details provided in the appendix represent valuable contributions to the field.

* The paper is well-structured, with clear presentation and thorough technical descriptions.

**Weaknesses:**

Major Concerns:

* The novelty and technical contribution appear limited. The core methodology seems to be primarily a combination of existing work (MaskGIT and SpearTTS). The speech acoustic code is derived from DAC, with no apparent novel methodological contributions.

* Regarding NAR methods, the authors cite phone-level duration prediction as a source of complex model design. However, this claim needs clarification: How is this complexity measured (e.g., training time, loss balancing)? Why are AR-based methods considered less complex? Experience with NaturalSpeech3/Fastspeech2 suggests duration modeling is manageable and straightforward to implement. Does this conclusion apply to traditional small-scale TTS models? The statement appears overly generalized.

* The Related Work section overlooks prominent flow-based methods like VoiceBox and p-flow, which also employ masked generative models. Their exclusion from this category requires explanation.

* Table 1 would be more appropriate in the appendix. Multi-task capability alone is not a significant contribution, as multi-task frameworks are common. The focus should be on concrete improvements in naturalness, prosody, efficiency, or reduced complexity. Placing Table 1 prominently suggests multi-task functionality is the main contribution, which diminishes the paper's actual value.

* In line 210, the claim that k-means based discrete SSL units lead to information loss needs substantiation. Why is VQ-VAE of W2V-BERT considered superior for semantic preservation despite also being discrete SSL units? While Appendix B provides semantic and acoustic comparisons, direct evidence supporting this specific claim is lacking.

Minor Points:

* Figure 2 would benefit from additional notations for clarity. Also, what is the difference between prompt semantic tokens and target semantic tokens? Can you give an example?

* While following the text-semantics-acoustics pipeline established by papers like SpearTTS, the superiority of this approach over alternatives (e.g., direct text2acoustic without semantics) should be justified.

* Section 4.2.3's purpose and motivation need clarification, particularly if addressing speed modification, as specialized tools exist for this purpose.

* Consider whether inference steps alone adequately measure efficiency, or if other metrics common in streaming methods (e.g., latency) would provide more comprehensive evaluation."

**Questions:**

See weaknesses.

---

> ### Author Response · Authors · 2024-11-17
> **Reply to Reviewer n456 (Part 1)**
>
> We sincerely thank the reviewer for their thorough review and valuable constructive feedback. We are also grateful for your recognition of our work's impact on the research community. We address each of your concerns in detail below.
>
> ## Weaknesses
>
> ``Weakness 1: Limited novelty as the method mainly combines existing works (MaskGIT, SpearTTS) with no apparent novel contributions in acoustic modeling (using DAC).``
>
> We would like to highlight MaskGCT's significant contributions at both representation and modeling levels: (1) At the modeling level, MaskGCT innovatively applies masked generative modeling to both text-to-semantic and semantic-to-acoustic generation. While this approach was originally developed for image generation (MaskGIT), its adaptation to TTS presents unique challenges and opportunities. Notably, MaskGCT achieves non-autoregressive TTS without requiring explicit phone-speech alignment supervision or phone-level duration prediction, which significantly simplifies the traditional TTS pipeline while achieving more natural and consistent generation results.
> (2) At the representation level, MaskGCT introduces VQ-based semantic tokens, which represents a distinct approach compared to previous methods that relied on k-means clustering.
>
> ``Weakness 2: The claim about NAR methods being complex due to duration prediction needs better justification, as duration modeling in models like NaturalSpeech3/FastSpeech2 is relatively straightforward.``
>
> We would like to clarify that the complexity of phone-level duration prediction arises from several key aspects:
>
> 1. Implementation Complexity: Phone-level duration prediction requires pre-extracted phone-level durations as supervision and additional modeling modules, which becomes more challenging in zero-shot scenarios and with in-the-wild or noisy data. For instance, NaturalSpeech 3 employs a Conformer-based duration predictor with discrete diffusion, while [1] relies on diffusion models.
>
> 2. Computational Overhead: The lack of efficient tools for phone-level duration extraction makes it computationally intensive, especially when processing large-scale datasets.
>
> 3. Accuracy Limitations: Phone-level duration extraction from in-the-wild data often lacks accuracy, potentially compromising model performance. Evidence from [2] shows that VoiceBox's reproduction on the Emilia dataset, which depends on phone-level duration prediction, achieved lower naturalness and similarity scores compared to autoregressive models that avoid such prediction.
>
> 4. Scalability Considerations: Our approach aims to reduce dependence on such priors while leveraging more powerful generative models and larger datasets to enhance generation quality. We acknowledge this may not fully apply to traditional small-data TTS scenarios, where phone-level duration prediction might be necessary for model convergence, as demonstrated in flow-matching-based works [3,4]. To validate our approach, we conducted additional experiments training T2S models on smaller datasets (LibriTTS and a 1K-hour Emilia subset) while maintaining the self-supervised S2A model. Results demonstrate robust performance even with limited training data, which we attribute to the effectiveness of semantic token prediction and the strong modeling capabilities of masked generative approaches.
>
> | Model | SIM-O ↑ | WER ↓ |
> |-------|----------|---------|
> || **SeedTTS *test-en*** ||
> | MaskGCT (LibriTTS 0.58K hours) | 0.677 | 3.043 |
> | MaskGCT (Emilia 1K hours) | 0.696 | 3.378 |
> | MaskGCT (Emilia 100K hours) | 0.728 | 2.466 |
> || **SeedTTS *test-zh*** ||
> | MaskGCT (Emilia 1K hours) | 0.754 | 3.012 |
> | MaskGCT (Emilia 100K hours) | 0.777 | 2.183 |
>
> ``Weakness 3: The Related Work section omits important flow-based methods (VoiceBox, p-flow) that use masked generative models.``
>
> We would like to note that VoiceBox was introduced in the first paragraph of our Related Work section and served as a baseline in our experimental comparisons. We acknowledge the oversight in not including the p-flow paper and will incorporate it in the revised version.
>
> Additionally, we want to clarify some potential misconception. VoiceBox is a flow matching-based TTS system, and its mask is designed to serve as a prompt during training, which is different from the "mask" in masked generative models. We believe the "mask" in masked generative models is more akin to the noise in diffusion models, as masked generative models have a discrete diffusion perspective.
>
> ``Weakness 4: Table 1's prominence suggests multi-task capability as the main contribution, while the focus should be on concrete improvements in performance and efficiency.``
>
> Thank you for your suggestions! We will reorganize it in the revised version of the paper.

---

> ### Author Response · Authors · 2024-11-17
> **Reply to Reviewer n456 (Part 2)**
>
> ``Weakness 5: The claim about information loss in k-means-based discrete SSL units compared to VQ-VAE needs better substantiation.``
>
> Thank you for your valuable suggestion. This question has also been raised by other reviewers, and we believe a thorough investigation of this matter will significantly strengthen our paper. We will first present our empirical findings (which are already included in our paper), followed by additional experimental results. **All these analyses will be incorporated into the revised version of our paper.**
>
> In our initial experiments, we observed that k-means-based semantic tokens were less effective in predicting acoustic tokens for languages with rich prosodic variations, particularly Chinese, where significant pitch variations were observed. We further support this finding with qualitative analysis.
>
> 1. Since k-means can be seen as optimizing the reconstruction loss between input and reconstruction, we present reconstruction loss curves under different k-means and VQ configurations.
> We compare four configurations: VQ-8192 (which is the same as in our paper), VQ-2048, k-means-8192, and k-means-2048.
> The loss curves can be found at this [link](https://github.com/maskgct/maskgct/raw/refs/heads/main/recon_loss_for_vq.PNG).
>
> 2. The information preservation in semantic tokens directly affects the semantic-to-acoustic model's prediction performance. We present the top-10 accuracy (shown in this [link](https://raw.githubusercontent.com/maskgct/maskgct/refs/heads/main/soundstorm_layer1_acc.PNG)) of the semantic-to-acoustic model in predicting the first layer of acoustic tokens. The results demonstrate that VQ-8192 outperforms VQ-2048, which in turn outperforms k-means-8192.
>
> 3. We investigate the impact of different semantic representation approaches on acoustic token reconstruction. We train separate semantic-to-acoustic (S2A) models for each configuration and evaluate their performance through speech reconstruction metrics. Across all three test sets, the results reveal a consistent performance ranking in similarity (SIM) scores, with VQ-8192 yielding the highest performance, followed sequentially by VQ-2048, k-means-8192, and k-means-2048. For WER, VQ-based methods also demonstrate superior performance over k-means approaches, though this advantage is less pronounced on LibriSpeech test-clean. Notably, on SeedTTS test-zh (Chinese test set), k-means exhibits a substantial degradation in WER. We attribute this to the stronger coupling between Chinese semantics and prosodic features, where the transition from VQ to k-means results in significant loss of prosodic information in the semantic representations.
>
> | Semantic Codec | LibriSpeech *test-clean* | | SeedTTS *test-en* | | SeedTTS *test-zh* | |
> |--------------|----------|---------|----------|---------|----------|---------|
> | | SIM-O ↑ | WER ↓ | SIM-O ↑ | WER ↓ | SIM-O ↑ | WER ↓ |
> | k-means 2048 | 0.648 | 3.013 | 0.658 | 3.989 | 0.691 | 11.420 |
> | k-means 8192 | 0.661 | 2.862 | 0.664 | 3.012 | 0.713 | 8.782 |
> | VQ 2048 | *0.671* | **2.177** | *0.692* | *3.187* | *0.744* | *4.913* |
> | VQ 8192 | **0.680** | *2.223* | **0.713** | **2.175** | **0.763** | **2.088** |
>
> We believe these additions will strengthen our argument and offer a clearer understanding of the impact of different semantic token approaches on model performance.
>
> ``Minor Point 1:  Figure 2 would benefit from additional notations for clarity. Also, what is the difference between prompt semantic tokens and target semantic tokens? Can you give an example?``
>
> The prompt serves as a reference for style, prosody, and speaker characteristics, enabling zero-shot TTS through in-context learning. During training, we randomly select a prefix from each sample as the prompt, keeping its semantic tokens unmasked. The model learns to predict masked tokens by conditioning on both the prompt's semantic tokens and the input text. For inference, we construct the input sequence by concatenating [prompt text, target text, prompt semantic tokens], which guides the model in predicting the target tokens.

---

> ### Author Response · Authors · 2024-11-17
> **Reply to Reviewer n456 (Part 3)**
>
> ``Minor Point 2: The advantage of using semantic tokens as an intermediate representation needs justification (e.g., direct text2acoustic without semantics) should be justified.``
>
> Thank you for this constructive suggestion. We have investigated this issue thoroughly and found several key advantages of our semantic token approach:
>
> 1. Performance Superiority: Recent works attempting to model continuous features like Mel-spectrograms directly without phone-level duration [3] show lower similarity compared to MaskGCT. Our experiments demonstrate that directly mapping text to multi-layer acoustic tokens using masked generative models leads to convergence difficulties, resulting in poor intelligibility and high WER.
>
> 2. Training Stability: Both [3,5] report convergence issues on small datasets when bypassing intermediate representations. This aligns with our early experimental findings.
>
> 3. Empirical Evidence: We conducted a comparative study between MaskGCT and a direct text-to-acoustic approach (implemented by removing semantic token conditioning and adding text conditioning to our semantic-to-acoustic model) on a 10K-hour subset. The results clearly show that direct acoustic token prediction from text faces convergence challenges, yielding lower SIM scores and substantially higher WER. This demonstrates that our two-stage approach effectively reduces the overall modeling complexity.
>
> We are currently extending these experiments to the full dataset and will include comprehensive results in the revised paper.
>
> | Model | SIM-O ↑ | WER ↓ |
> |-------|----------|---------|
> ||**SeedTTS *test-en***|
> | Text-to-Acoustic (Emilia 10K hours) | 0.651 | 12.75 |
> | MaskGCT (Emilia 10K hours) | **0.719** | **2.872** |
> || **SeedTTS *test-zh*** |
> | Text-to-Acoustic (Emilia 10K hours) | 0.727 | 17.08 |
> | MaskGCT (Emilia 10K hours) | **0.762** | **3.302** |
>
> ``Minor Point 3:  Section 4.2.3's purpose and motivation need clarification, particularly if addressing speed modification, as specialized tools exist for this purpose.``
>
> This section aims to demonstrate two key aspects of our model:
>
> 1. Generation Capability: MaskGCT can produce high-quality speech across a wide range of durations while maintaining natural prosody. This is fundamentally different from simple speech rate adjustment, which often introduces artifacts in prosody, pitch, and timbre.
>
> 2. Control and Diversity: Our approach offers better controllability over speech generation compared to AR models, while ensuring diverse outputs. We have provided audio samples on our demo page that showcase natural prosodic variations across different durations.
>
> ``Minor Point 4: Additional efficiency metrics beyond inference steps (e.g., latency) should be considered.``
>
> Thanks for your suggestion. We provide the real-time factor (RTF) of MaskGCT on an A100 GPU for generating a 20-second speech across various inference steps in Table. Across all configurations presented, there is no significant performance difference. Additionally, we also present the RTF of AR + SoundStorm. For AR + SoundStorm, generating a 20-second speech requires 20 * 50 = 1000 steps for text-to-semantic inference. However, we can leverage kv-cache to accelerate the process.
>
> | Model | T2S steps | S2A steps | RTF |
> |-------|-----------|-----------|-----|
> | MaskGCT | 50 | [40, 16, 1, 1, 1, 1, 1, 1, 1, 1, 1, 1] | 0.52 |
> | MaskGCT | 50 | [10, 1, 1, 1, 1, 1, 1, 1, 1, 1, 1, 1] | 0.44 |
> | MaskGCT | 25 | [10, 1, 1, 1, 1, 1, 1, 1, 1, 1, 1, 1] | 0.31 |
> | AR + SoundStorm | 1000 | [40, 16, 1, 1, 1, 1, 1, 1, 1, 1, 1, 1] | 0.98 |
>
> Thanks again for your constructive comments. We would be grateful if we could hear your feedback regarding our answers to the reviews. We would be happy to answer and discuss if you have further comments.
>
> [1] Li X, Liu S, Lam M W Y, et al. Diverse and expressive speech prosody prediction with denoising diffusion probabilistic model[J]. arXiv preprint arXiv:2305.16749, 2023.
>
> [2] He H, Shang Z, Wang C, et al. Emilia: An extensive, multilingual, and diverse speech dataset for large-scale speech generation[J]. arXiv preprint arXiv:2407.05361, 2024.
>
> [3] Chen Y, Niu Z, Ma Z, et al. F5-TTS: A Fairytaler that Fakes Fluent and Faithful Speech with Flow Matching[J]. arXiv preprint arXiv:2410.06885, 2024.
>
> [4] A^2-Flow: Alignment-Aware Pre-training for Speech Synthesis with Flow Matching https://openreview.net/attachment?id=e2p1BWR3vq&name=pdf

---

> > ### Author Response · Authors · 2024-11-20
> >
> > Thanks again for your valuable comments, we would be grateful if we could hear your feedback regarding our answers. We would be happy to answer and discuss if you have further comments.

---

> > > ### Author Response · Authors · 2024-11-22
> > >
> > > Thank you again for your great efforts and the valuable comments.
> > >
> > > We have carefully addressed the main concerns in detail. We hope you might find the response satisfactory. As the paper discussion phase is coming to an end, we would be grateful if we could hear your feedback regarding our answers to the reviews. We will be very happy to clarify any remaining points (if any).
> > >
> > > We sincerely hope you can consider increasing the score if you find our reply solves your concerns.

---

> > > > ### Comment · Reviewer_n456 · 2024-11-23
> > > >
> > > > 1. For novelty.
> > > >
> > > > The authors state "While this approach was originally developed for image generation (MaskGIT), its adaptation to TTS presents unique challenges and opportunities." “Adaptation” reflects the limited novelty of the work. While MaskGCT introduces VQ-based semantic tokens, this seems to be more of an incremental modification better suited for ablation studies. Overall, there appears to be no significant differentiation between this work and SpearTTS. The method is an A+B style work, which other reviewers have also noted as a concern. For instance, we could take an existing paper like VoiceBox and replace one of its modules with a new model proposed in computer vision to potentially achieve improvements. While the engineering contribution is acknowledged, the novelty contribution is really poor.
> > > >
> > > > 2. Regarding Duration Modeling:
> > > >
> > > > In TTS, we primarily evaluate zero-shot inference results, which don't require pre-extracted phoneme duration. Therefore, this isn't necessarily a drawback in NaturalSpeech3. While the authors mention "the lack of efficient tools for phone-level duration extraction," using MFA for offline extraction, despite not being 100% accurate, remains a viable approach. Furthermore, it's unclear whether the improvement in naturalness stems from the removal of duration prediction, as noted in paper [2]. If the authors intend to demonstrate that duration-based methods are inferior to AR-based methods for large-scale TTS work, additional metrics and experiments are needed.
> > > >
> > > > For example, replacing AR modules with duration-based methods would provide direct performance comparisons. Additionally, you should analyze how AR models implicitly handle speech-text alignment and develop metrics to evaluate this alignment quality (either via ground truth or phoneme-based tasks). A key question to address is: Do AR-based methods perform better simply because they achieve better alignment, or are other factors at play? The current work doesn't fully explain why duration models perform worse, making it difficult to understand the true advantages of AR-based approaches. Understanding these mechanisms would not only validate the authors' claims but also provide valuable guidance for future TTS system design.
> > > >
> > > > 3. VQ semantics vs Kmeans semantics.
> > > >
> > > > The reconstruction loss and semantic-to-acoustic generation metrics are not sufficient evidence to conclude that VQ contains more semantic information than K-means. These metrics primarily measure global patterns, while semantics are fundamentally local patterns. The definition of semantics used in this work needs clarification. Semantics cannot be reduced to good reconstruction quality or high WER scores in speech recognition. Rather, semantics refers to the clear delineation of linguistic units such as words, syllables, and phoneme boundaries. Local metrics should be developed to properly evaluate these semantic properties. For better understanding of semantics, please refer to the DINO paper (https://arxiv.org/abs/2104.14294). Additional relevant references in speech include:
> > > >
> > > > https://arxiv.org/abs/2305.10005
> > > > https://arxiv.org/abs/2310.10803
> > > >
> > > > Overall, my concerns have not been adequately addressed. The paper would be substantially stronger with more rigorous statements, detailed explanations, and the development of appropriate metrics.

---

> > > > > ### Author Response · Authors · 2024-11-27
> > > > >
> > > > > Thanks for your comments. Here are our replies for your remaining concerns.
> > > > >
> > > > > **novelty**
> > > > >
> > > > > Masked generative models have been proven to be a class of versatile and powerful generative models, like diffusion models, which have been applied in various fields such as images, videos, and audio. However, how to leverage masked generative models for better TTS is worth exploring. Previous works, such as SoundStorm and NaturalSpeech 3, have also used masked generative modeling, but they both require frame-level conditioning: NaturalSpeech 3 relies on traditional phone-level duration predictors, while SoundStorm is based on AR. A key question MaskGCT explores is whether NAR TTS models can eliminate the dependency on phone-level duration supervision. Additionally, MaskGCT investigates VQ-based semantic tokens, and in our supplementary experiments, we also explore the differences between two-stage models and direct text-to-acoustic modeling with masked generative modeling.
> > > > >
> > > > > We also illustrate the differences between MaskGCT and traditional NAR (Non-Autoregressive) methods in text-speech alignment modeling through a figure (in this [https://raw.githubusercontent.com/maskgct/maskgct/refs/heads/main/aatn_map.png](https://raw.githubusercontent.com/maskgct/maskgct/refs/heads/main/aatn_map.png)). For traditional NAR TTS systems, a duration predictor and length regulator are first used to obtain frame-level conditions that provide alignment information. For MaskGCT, similar to AR (Autoregressive) methods, we concatenate the text in front of the speech tokens through in-context learning, and the model implicitly learns speech-text alignment through self-attention.
> > > > >
> > > > > **Regarding Duration Modeling**
> > > > >
> > > > > For models like NaturalSpeech 3 and VoiceBox that rely on in-context learning with a prompt mechanism, extracting the duration of prompt phones is required during inference. While we do not claim that this is necessarily a drawback in NaturalSpeech 3, it does introduce prior knowledge and additional modules, which are worth exploring to determine if they can be eliminated. I believe eliminating more priors and redundant modules is not only a development direction in the TTS field but also a general trend in most domains. The experiment results show that our method outperforms or matches previous AR and NAR (which require phone-level duration) in terms of metrics. Compared to NAR models that require phone-level duration, our method is simpler, does not need phone-level duration, and achieves better SIM, CMOS, and SMOS. Compared to AR models that also do not require phone-level duration, our method has fewer inference steps, is faster, and achieves better SIM, WER, CMOS, and SMOS. In addition, we provide attention maps at different stages of inference (steps 1, 10, 20) and across various layers of the model (layer 1, layer 6, layer 16) in this [https://raw.githubusercontent.com/maskgct/maskgct/refs/heads/main/ar_vs_nar_vs_maskgct.png](https://raw.githubusercontent.com/maskgct/maskgct/refs/heads/main/ar_vs_nar_vs_maskgct.png). These attention maps demonstrate that our model implicitly learns the alignment between text and speech. It shows that the model handles the alignment between speech tokens and text in the middle layers of the network, while in the deeper layers, the model has already resolved the alignment and focuses on processing the tokens.

---

> > > > > > ### Author Response · Authors · 2024-11-27
> > > > > >
> > > > > > **VQ semantics vs Kmeans semantics**
> > > > > >
> > > > > > For the third point, in our paper, our main focus is on how semantic tokens obtained through better discretization methods can better aid in predicting acoustic tokens to reconstruct high-quality speech waveforms. The reconstruction loss of different discretization methods (VQ vs. K-means) supports our claim that "This approach minimizes the information loss of semantic features even with a single codebook." Moreover, semantic tokens based on VQ actually lead to better speech similarity and WER, which are reasonable metrics for evaluating zero-shot TTS. As for the description of semantic features, we discuss this in the appendix: "In this paper, we refer to the speech representation extracted from the speech self-supervised learning (SSL) model as the semantic feature. The discrete tokens obtained through the discretization of these semantic features (using k-means or vector quantization) are termed semantic tokens. Similarly, we define the representations from melspectrogram, neural speech codecs, or speech VAE as acoustic features, and their discrete counterparts are called acoustic tokens. This terminology was first introduced in [68] and has since been adopted by many subsequent works [8, 19, 39, 69, 70]. It is important to note that this is not a strictly rigorous definition. Generally, we consider semantic features or tokens to contain more prominent linguistic information and exhibit stronger correlations with phonemes or text. One measure of this is the phonetic discriminability in terms of the ABX error rate. In this paper, the W2v-BERT 2.0 features we use have a phonetic discriminability within less than 5 on the LibriSpeech dev-clean dataset, whereas acoustic features, for example, Encodec latent features, score above 20 on this metric. However, it is worth noting that semantic features or tokens not only contain semantic information but also include prosodic and timbre aspects. In fact, we suggest that for certain two-stage zero-shot TTS systems, excessive loss of information in semantic tokens can degrade the performance of the second stage, where semantic-to-acoustic conversion occurs. Therefore, finding a speech representation that is more suitable for speech generation remains a challenging problem."
> > > > > >
> > > > > > [8] Naturalspeech 3: Zero-shot speech synthesis with factorized codec and diffusion models. arXiv preprint arXiv:2403.03100, 2024.
> > > > > > [19] Soundstorm: Efficient parallel audio generation. arXiv preprint arXiv:2305.09636, 2023.
> > > > > > [39] Repcodec: A speech representation codec for speech tokenization. arXiv preprint arXiv:2309.00169, 2023.
> > > > > > [68] Audiolm: a language modeling approach to audio generation. IEEE/ACM transactions on audio, speech, and language processing, 31:2523–2533, 2023.
> > > > > > [69] Fireredtts: A foundation text-to-speech framework for industry-level generative speech applications. arXiv preprint arXiv:2409.03283, 2024.
> > > > > > [70] Speechtokenizer: Unified speech tokenizer for speech large language models. arXiv preprint arXiv:2308.16692, 2023.

---

> > > > > > > ### Author Response · Authors · 2024-11-27
> > > > > > >
> > > > > > > Thanks again for your feedback. We would like to know if there are any further questions or issues we can address. We are committed to engaging fully and will reply as promptly as possible. If you have any remaining concerns or suggestions to help further enhance this research, please feel free to comment, and we are more than willing to continue addressing these issues.

---

> > > > > > > > ### Author Response · Authors · 2024-12-03
> > > > > > > >
> > > > > > > > Thanks again for your feedback. Since the rebuttal phase is nearing its end, we would appreciate it if you could let us know whether our further reply has addressed your concern. If you have any further suggestions or concerns, we are more than willing to provide more discussion as soon as possible.

---

### Official Review · Reviewer_QWox · 2024-11-04

**Soundness:** 3
**Presentation:** 3
**Contribution:** 3
**Rating:** 6
**Confidence:** 5

**Summary:**

This paper proposes a masked generative codec transformer, MaskGCT, which performs speech synthesis using a mask-predict approach. MaskGCT generates semantic tokens from text input and is composed of two components: text-to-semantic MaskGCT, which generates semantic tokens from text, and semantic-to-acoustic MaskGCT, which generates acoustic tokens from semantic tokens. Each MaskGCT operates by filling masked parts based on confidence scores predicted in stages, from a sequence where all discrete speech tokens are masked at the start. Instead of using the commonly used method of obtaining semantic tokens through k-means clustering applied to pre-trained SSL models, MaskGCT employs a separate VQ-VAE model on SSL features to generate semantic tokens, reducing information loss through reconstruction loss. MaskGCT uses a similar mask prediction method for predicting discrete speech tokens as Soundstorm, which is based on MaskGIT. However, unlike Soundstorm, MaskGCT approaches the semantic token generation from text through masked token modeling and includes an additional module for total length prediction to align sequence lengths. MaskGCT demonstrates superior speaker similarity in widely used zero-shot TTS evaluation compared to various zero-shot TTS models. Moreover, a demo page with samples shows the versatility of MaskGCT across various tasks.

**Strengths:**

* The paper shows that a mask-predict approach on discrete tokens can achieve a high level of speaker similarity.

* Samples on the demo page show that MaskGCT enables zero-shot TTS that effectively captures not only timbre but also emotion and style across a variety of voices, with excellent sample quality.

* To assess how well the model captures accent and emotion, the paper introduces metrics such as Accent SIM and Emotion SIM, which utilize representations reflecting each attribute. Their results indicate that MaskGCT performs better in imitating accent and emotion than other approaches based on these metrics.

**Weaknesses:**

* In Soundstorm, the mask-predict approach is introduced to acoustic token generation. It seems that the methodological novelty in MaskGCT lies mainly in extending this mask-predict approach to semantic token generation.

* The paper utilizes VQ-VAE to reduce information loss in semantic tokens compared to the k-means clustering approach; however, it does not experimentally demonstrate how this approach improves over k-means clustering.

* Additionally, while using a larger number of semantic tokens may yield good performance in zero-shot TTS, the semantic tokens do not seem disentangled from speaker information, making MaskGCT less suitable for voice conversion tasks. The voice conversion samples on the demo page also appear less similar to the reference speaker.

* When comparing single-sample generation, MaskGCT's pronunciation accuracy appears lower than that of VoiceBox or NaturalSpeech 3.

* Regarding the seed-TTS evaluation method, although it is not explicitly shown as a baseline in the paper, the objective metrics appear to be worse than those of seed-TTS.

**Questions:**

Comments

* It would be helpful to mention the frame rate of semantic and acoustic tokens.
* I am curious whether only a single sampling step is used from the third layer during acoustic token generation. Is there any difference when using more sampling steps?
* It would be beneficial to also present the overall sampling speed of the model, including the Real-Time Factor.
* In Table 1, it seems that VoiceBox's Imp. Dur. should be marked as "X."

---

> ### Author Response · Authors · 2024-11-17
> **Reply to Reviewer QWox (Part 1)**
>
> First of all, we want to thank the reviewer for your careful reading and providing a lot of constructive comments! Below we address the concerns mentioned in the review.
>
> ## Weaknesses
>
> ``Weakness 1: The main novelty appears to be extending Soundstorm's mask-predict approach from acoustic to semantic token generation.``
>
> MaskGCT employs masked generative modeling for both text-to-semantic and semantic-to-acoustic generation. Beyond this, we would like to highlight that MaskGCT's novelty lies in two key aspects: (1) MaskGCT explores non-autoregressive TTS without requiring explicit phone-speech alignment supervision or phone-level duration prediction, which significantly simplifies the traditional TTS pipeline while achieving more natural and consistent generation results. (2) MaskGCT introduces VQ-based semantic tokens, which represents a distinct approach compared to previous methods that relied on k-means clustering.
>
> ``Weakness 2: The paper lacks experimental comparison between the proposed VQ-VAE and k-means clustering for semantic tokens``
>
> Thank you for your valuable suggestion. This question has also been raised by other reviewers, and we believe a thorough investigation of this matter will significantly strengthen our paper. We will first present our empirical findings (which are already included in our paper), followed by additional experimental results. **All these analyses will be incorporated into the revised version of our paper.**
>
> In our initial experiments, we observed that k-means-based semantic tokens were less effective in predicting acoustic tokens for languages with rich prosodic variations, particularly Chinese, where significant pitch variations were observed. We further support this finding with qualitative analysis.
>
> 1. Since k-means can be seen as optimizing the reconstruction loss between input and reconstruction, we present reconstruction loss curves under different k-means and VQ configurations.
> We compare four configurations: VQ-8192 (which is the same as in our paper), VQ-2048, k-means-8192, and k-means-2048.
> The loss curves can be found at this [link](https://github.com/maskgct/maskgct/raw/refs/heads/main/recon_loss_for_vq.PNG).
>
> 2. The information preservation in semantic tokens directly affects the semantic-to-acoustic model's prediction performance. We present the top-10 accuracy (shown in this [link](https://raw.githubusercontent.com/maskgct/maskgct/refs/heads/main/soundstorm_layer1_acc.PNG)) of the semantic-to-acoustic model in predicting the first layer of acoustic tokens. The results demonstrate that VQ-8192 outperforms VQ-2048, which in turn outperforms k-means-8192.
>
> 3. We investigate the impact of different semantic representation approaches on acoustic token reconstruction. We train separate semantic-to-acoustic (S2A) models for each configuration and evaluate their performance through speech reconstruction metrics. Across all three test sets, the results reveal a consistent performance ranking in similarity (SIM) scores, with VQ-8192 yielding the highest performance, followed sequentially by VQ-2048, k-means-8192, and k-means-2048. For WER, VQ-based methods also demonstrate superior performance over k-means approaches, though this advantage is less pronounced on LibriSpeech test-clean. Notably, on SeedTTS test-zh (Chinese test set), k-means exhibits a substantial degradation in WER. We attribute this to the stronger coupling between Chinese semantics and prosodic features, where the transition from VQ to k-means results in significant loss of prosodic information in the semantic representations.
>
> | Semantic Codec | LibriSpeech *test-clean* | | SeedTTS *test-en* | | SeedTTS *test-zh* | |
> |--------------|----------|---------|----------|---------|----------|---------|
> | | SIM-O ↑ | WER ↓ | SIM-O ↑ | WER ↓ | SIM-O ↑ | WER ↓ |
> | k-means 2048 | 0.648 | 3.013 | 0.658 | 3.989 | 0.691 | 11.420 |
> | k-means 8192 | 0.661 | 2.862 | 0.664 | 3.012 | 0.713 | 8.782 |
> | VQ 2048 | *0.671* | **2.177** | *0.692* | *3.187* | *0.744* | *4.913* |
> | VQ 8192 | **0.680** | *2.223* | **0.713** | **2.175** | **0.763** | **2.088** |
>
> These comprehensive analyses demonstrate the advantages of VQ-based semantic tokens over k-means clustering, providing strong empirical evidence to support our design choices in MaskGCT.

---

> ### Author Response · Authors · 2024-11-17
> **Reply to Reviewer QWox (Part 2)**
>
> ``Weakness 3: The semantic tokens are not well disentangled from speaker information, making MaskGCT less effective for voice conversion tasks.``
>
> Since MaskGCT was primarily designed for Zero-Shot TTS tasks, we did not extensively focus on decoupling speaker information from semantic tokens. However, in our recent research, we have enabled voice conversion capabilities through a simpler approach in the semantic-to-acoustic model. Specifically, we utilize a lightweight yet efficient voice conversion model (such as OpenVoice), to perform real-time voice conversion on the target speech using randomly sampled prompt speech, thereby achieving timbre perturbation. These perturbed semantic tokens are then used as input to predict target acoustic tokens with the prompt in the semantic-to-acoustic model. The results are presented below. **We will incorporate these details in the revised version of the paper.**  We use the VCTK dataset to evaluate our system, randomly selecting 200 samples as source speech. For each sample, we randomly select another sample from the same speaker as the prompt speech.
>
> | Model | SIM-O(↑) | WER(↓) | DNSMOS(↑) | NISQA(↑) |
> |-------|-----------|---------|------------|-----------|
> | HireSpeech++ [1] | 0.379 | 4.87 | 3.402 | 3.794 |
> | LM-VC [2] | 0.286 | 8.35 | 3.457 | 3.927 |
> | UniAudio [3] | 0.249 | 9.00 | 3.472 | 4.279 |
> | MaskGCT-VC | **0.532** | **4.49** | **3.510** | **4.469** |
>
> ``Weakness 4: MaskGCT shows lower pronunciation accuracy compared to VoiceBox/NaturalSpeech 3 in single-sample generation.``
>
> I believe the reason MaskGCT outperforms NaturalSpeech 3 and VoiceBox in terms of WER metrics may be due to ASR's tendency to more accurately recognize standard speech. This is because NS3 and VoiceBox incorporate phone-level duration predictors, and the LibriSpeech dataset, being a more standardized recording test set, allows NS3 and VoiceBox to predict durations more effectively. However, for CMOS, MaskGCT performs better than VoiceBox.
>
> ``Weakness 5: The objective metrics appear to be worse than seed-TTS baseline.``
>
> As far as I know, Seed-TTS utilizes over 2 million hours of training data, which is 20 times the amount used by our model. Consequently, it is challenging for our model to achieve a fair comparison in performance with Seed-TTS.
>
> ## Questions
>
> ``Q1: It would be helpful to mention the frame rate of semantic and acoustic tokens.``
>
>
> As detailed in Appendix A.4, our model operates with two types of tokens: semantic tokens (16KHz sampling rate, 320 hopsize) and acoustic tokens (24KHz sampling rate, 480 hopsize). Both tokens effectively maintain a consistent frame rate of 50Hz. We will move this important technical detail to the main text for better clarity.
>
> ``Q2: I am curious whether only a single sampling step is used from the third layer during acoustic token generation. Is there any difference when using more sampling steps?``
>
> Given that the first layer of acoustic tokens carries the majority of information, we allocate more inference steps to this layer while reducing steps for subsequent layers. As demonstrated in Appendix A.3, we have analyzed how varying the number of inference steps in the second-stage model affects overall performance. Our experiments show that even a minimal configuration [10, 1, 1, 1, 1, 1, 1, 1, 1, 1, 1, 1] maintains comparable performance, with only marginal decreases in SIM and WER metrics.
>
> ``Q3: It would be beneficial to also present the overall sampling speed of the model, including the Real-Time Factor.``
>
> Thanks for your suggestion.  **We will also add it to the revised version of the paper.** We present the real-time factor (RTF) of MaskGCT on an A100 GPU for generating a 20-second speech across various inference steps in Table. Across all configurations presented, there is no significant performance difference. Additionally, we also present the RTF of AR + SoundStorm. For AR + SoundStorm, generating a 20-second speech requires 20 * 50 = 1000 steps for text-to-semantic inference. However, we can leverage kv-cache to accelerate the process.
>
> | Model | T2S steps | S2A steps | RTF |
> |-------|-----------|-----------|-----|
> | MaskGCT | 50 | [40, 16, 1, 1, 1, 1, 1, 1, 1, 1, 1, 1] | 0.52 |
> | MaskGCT | 50 | [10, 1, 1, 1, 1, 1, 1, 1, 1, 1, 1, 1] | 0.44 |
> | MaskGCT | 25 | [10, 1, 1, 1, 1, 1, 1, 1, 1, 1, 1, 1] | 0.31 |
> | AR + SoundStorm | 1000 | [40, 16, 1, 1, 1, 1, 1, 1, 1, 1, 1, 1] | 0.98 |

---

> ### Author Response · Authors · 2024-11-17
> **Reply to Reviewer QWox (Part 3)**
>
> ``Q4: In Table 1, it seems that VoiceBox's Imp. Dur. should be marked as "X."``
>
> Thank you for the reminder! We will fix it in the revised version.
>
> Thanks again for your constructive comments. We would be grateful if we could hear your feedback regarding our answers to the reviews. We would be happy to answer and discuss if you have further comments.
>
> [1] Lee S H, Choi H Y, Kim S B, et al. Hierspeech++: Bridging the gap between semantic and acoustic representation of speech by hierarchical variational inference for zero-shot speech synthesis[J]. arXiv preprint arXiv:2311.12454, 2023.
>
> [2] Wang Z, Chen Y, Xie L, et al. Lm-vc: Zero-shot voice conversion via speech generation based on language models[J]. IEEE Signal Processing Letters, 2023.
>
> [3] Yang D, Tian J, Tan X, et al. Uniaudio: An audio foundation model toward universal audio generation[J]. arXiv preprint arXiv:2310.00704, 2023.

---

> > ### Author Response · Authors · 2024-11-20
> >
> > Thanks again for your valuable comments, we would be grateful if we could hear your feedback regarding our answers. We would be happy to answer and discuss if you have further comments.

---

> > > ### Author Response · Authors · 2024-11-22
> > >
> > > Thank you again for your great efforts and the valuable comments.
> > >
> > > We have carefully addressed the main concerns in detail. We hope you might find the response satisfactory. As the paper discussion phase is coming to an end, we would be grateful if we could hear your feedback regarding our answers to the reviews. We will be very happy to clarify any remaining points (if any).

---

> > > > ### Comment · Reviewer_QWox · 2024-11-28
> > > >
> > > > The authors have made a lot of effort to address various comments, and I appreciate their work in resolving them. For weakness 2, they have clearly demonstrated that the VQ-VAE approach is superior to the k-means approach from the TTS perspective, which was previously overlooked. I also confirmed that updates related to voice conversion have been made in the paper, and it seems that weakness 5 and other questions have also been adequately addressed.
> > > >
> > > > However, regarding weakness 4, I am skeptical that the reason for MaskGCT's poor performance is that ASR aligns more easily with standard speech. When the ground truth length is used, MaskGCT also performs well, suggesting that the issue might not lie in ASR's tendencies but rather in occasional failures of the text-to-semantic token conversion. Regardless of the exact cause, based on the results presented in the paper and the rebuttal, I intend to maintain my score at 6.

---

> > > > > ### Author Response · Authors · 2024-11-28
> > > > >
> > > > > Dear Reviewer,
> > > > >
> > > > > Thank you for your feedback. We greatly appreciate your recognition of our responses. Regarding Weakness 4, our experiments on accent and emotion were designed to demonstrate the capabilities of MaskGCT in style imitation. The results show that MaskGCT's CMOS, SMOS, Accent (Emotion) MOS all surpass all baselines. For WER, in fact, our experiments indicate that the ground truth speech on these two datasets also has a higher WER than the previous results, which are 10.90 and 11.79, respectively. The results of MaskGCT (6.382 and 12.502) show no significant difference compared to the ground truth, and are very close to the best results among all baseline systems.
> > > > >
> > > > > If you have more suggestions to help us improve the paper, please feel free to let us know. We are more than willing to address any further concerns you may have. Thank you very much!

---

### Author Response · Authors · 2024-11-17

First of all, we would like to express our gratitude to all the reviewers for their invaluable feedback, which has significantly contributed to enhancing the quality of our paper. Here, we summarize some points raised by multiple reviewers. For each reviewer, we address their specific concerns in the respective comments section.

## The novelty of MaskGCT

We sincerely appreciate multiple reviewers' recognition of our engineering contributions. We have provided detailed model architectures and training specifications in the appendix, and we commit to open-sourcing all checkpoints and code to further benefit the research community.

MaskGCT represents the first NAR TTS system trained on large-scale in-the-wild datasets without phone-level duration prediction. Notably, it achieves state-of-the-art performance across multiple evaluation metrics on various test sets, demonstrating superior capabilities in multilingual generation, accent handling, and emotional expression.

In addition, we also highlight that MaskGCT's novelty lies in two key aspects: (1) MaskGCT explores non-autoregressive TTS without requiring explicit phone-speech alignment supervision or phone-level duration prediction, which significantly simplifies the traditional TTS pipeline while achieving more natural and consistent generation results. (2) MaskGCT introduces VQ-based semantic tokens, which represents a distinct approach compared to previous methods that relied on k-means clustering.

## The choice of semantic codec

All the reviewers have raised concerns about the choice of semantic codec (VQ vs. k-means). We believe a thorough investigation of this matter will significantly strengthen our paper. We will first present our empirical findings (which are already included in our paper), followed by additional experimental results. **All these analyses will be incorporated into the revised version of our paper.**

In our initial experiments, we observed that k-means-based semantic tokens were less effective in predicting acoustic tokens for languages with rich prosodic variations, particularly Chinese, where significant pitch variations were observed. We further support this finding with qualitative analysis.

1. Since k-means can be seen as optimizing the reconstruction loss between input and reconstruction, we present reconstruction loss curves under different k-means and VQ configurations.
We compare four configurations: VQ-8192 (which is the same as in our paper), VQ-2048, k-means-8192, and k-means-2048.
The loss curves can be found at this [link](https://github.com/maskgct/maskgct/raw/refs/heads/main/recon_loss_for_vq.PNG).

1. The information preservation in semantic tokens directly affects the semantic-to-acoustic model's prediction performance. We present the top-10 accuracy (shown in this [link](https://raw.githubusercontent.com/maskgct/maskgct/refs/heads/main/soundstorm_layer1_acc.PNG)) of the semantic-to-acoustic model in predicting the first layer of acoustic tokens. The results demonstrate that VQ-8192 outperforms VQ-2048, which in turn outperforms k-means-8192.

2. We investigate the impact of different semantic representation approaches on acoustic token reconstruction. We train separate semantic-to-acoustic (S2A) models for each configuration and evaluate their performance through speech reconstruction metrics. Across all three test sets, the results reveal a consistent performance ranking in similarity (SIM) scores, with VQ-8192 yielding the highest performance, followed sequentially by VQ-2048, k-means-8192, and k-means-2048. For WER, VQ-based methods also demonstrate superior performance over k-means approaches, though this advantage is less pronounced on LibriSpeech test-clean. Notably, on SeedTTS test-zh (Chinese test set), k-means exhibits a substantial degradation in WER. We attribute this to the stronger coupling between Chinese semantics and prosodic features, where the transition from VQ to k-means results in significant loss of prosodic information in the semantic representations.

| Semantic Codec | LibriSpeech *test-clean* | | SeedTTS *test-en* | | SeedTTS *test-zh* | |
|--------------|----------|---------|----------|---------|----------|---------|
| | SIM-O ↑ | WER ↓ | SIM-O ↑ | WER ↓ | SIM-O ↑ | WER ↓ |
| k-means 2048 | 0.648 | 3.013 | 0.658 | 3.989 | 0.691 | 11.420 |
| k-means 8192 | 0.661 | 2.862 | 0.664 | 3.012 | 0.713 | 8.782 |
| VQ 2048 | *0.671* | **2.177** | *0.692* | *3.187* | *0.744* | *4.913* |
| VQ 8192 | **0.680** | *2.223* | **0.713** | **2.175** | **0.763** | **2.088** |

We believe these additions will strengthen our argument and offer a clearer understanding of the impact of different semantic token approaches on model performance.

---

> ### Author Response · Authors · 2024-11-17
> **Paper Update**
>
> ## Additional Comparisons
>
> We have conducted extensive comparative experiments to provide a more comprehensive understanding of MaskGCT:
>
> 1. Efficiency Analysis: Compared RTF (Real-Time Factor) between MaskGCT under various inference parameters and AR models
>
> 2. Small Dataset Performance: Evaluated MaskGCT's effectiveness when trained on limited data
>
> 3. Architecture Comparison: Conducted comparative studies between MaskGCT and direct text-to-acoustic models
>
> 4. Voice Conversion Enhancement: Introduced a novel approach to improve MaskGCT's voice conversion capabilities, with comparative evaluations against baseline methods
>
>
> ## Revised Paper
>
> In addition, **we have uploaded a revised version of the paper with all changes highlighted in blue**. The key updates include:
>
>  - Fixed a typo in the system comparison figure and moved it to the Appendix (page 20, start at line 1050)
>
> - Added P-Flow reference in the related work section (page 2, line 101)
>
>  - Added analysis of semantic representation codecs (VQ vs. k-means, vocabulary size) in Section 4.4 (page 9, start at line 466)
>
> - Included RTF measurements under different settings in Appendix A.3 (page 13, start at line 938)
>
> - Added performance comparison on small datasets in Appendix K: Discussion about Concurrent Works (page 24, start at line 1290)
>
> - Added performance comparison between MaskGCT and direct text-to-acoustic model in Appendix K: Discussion about Concurrent Works (page 25, start at line 1307)
>
> - Added new voice conversion improvements in Appendix I: Voice Conversion (page 23, start at line 1236)
>
> - Included explanation for WER variations in Section 4.3: Speech Style Imitation (page 9, start at line 441)
>
> - Detailed two approaches for total duration calculation in Appendix A.5 (page 19, start at line 1025)
>
> All modifications are clearly marked in the revised manuscript. We welcome any additional questions or feedback.

---

### Meta-Review · Area_Chair_G2K9 · 2024-12-08

**Metareview:**

The paper introduces MaskGCT, a two-stage NAR TTS system that predicts semantic tokens from text and then acoustic tokens from semantic tokens using a masked generative transformer. MaskGCT demonstrates superior speaker similarity in widely used zero-shot TTS evaluation compared to various zero-shot TTS models. The main contribution of this paper is to show that it is possible to train a NAR TTS model with a simple pipeline without requiring explicit phone-speech alignment.

**Additional Comments On Reviewer Discussion:**

Most reviewers (QWox, AvMh, EzBH) maintained their scores around the acceptance threshold (6), acknowledging the authors’ rebuttals addressed several points and improved clarity. Reviewer n456 remained unconvinced on novelty, keeping a low score (3).

All reviewers raised concerns about the novelty of the work. Reviewer QWox noted that the mask-predict approach is applied to acoustic token generation, with the primary methodological novelty of MaskGCT being its extension of this approach to semantic token generation. Reviewer n456 commented that the novelty and technical contributions seem limited, as the core methodology largely combines existing work, specifically MaskGIT and SPEAR-TTS. Reviewer AvMh remarked that this work aligns with SPEAR-TTS and pointed out that the SoundStorm paper also explores a similar strategy. Furthermore, the application of a non-autoregressive masked generative transformer for codec-based TTS has already been presented in SoundStorm and NaturalSpeech3. Reviewer EzBH highlighted that using text as a condition without duration expansion for a NAR model was proposed in e2-TTS.

Author reply: The authors highlight the novelty with the following two points:  (1) MaskGCT explores non-autoregressive TTS without requiring explicit phone-speech alignment supervision or phone-level duration prediction, which significantly simplifies the traditional TTS pipeline while achieving more natural and consistent generation results. (2) MaskGCT introduces VQ-based semantic tokens, which represents a distinct approach compared to previous methods that relied on k-means clustering.

Note: Reviewer n456 is not convised

All reviewers have questions about utilising VQ-VAE to reduce information loss in semantic tokens compared to the k-means clustering approach.

Author reply: The authors show that the VQ-VAE approach is superior to the k-means approach from the TTS perspective.

Note: The author's reply addresses the issue.

AC's opinion:

The main consideration for whether to accept this paper lies in its novelty, which can be subjective. This paper can be seen as an integration of SPEAR-TTS (existing two-stage TTS methods) and MaskGIT (from computer vision). Although MaskGIT itself is not novel, I was genuinely surprised to discover that TTS can be trained using MaskGIT without requiring text and speech alignment. From this perspective, I am inclined to accept this paper.

However, a very recent work, E2-TTS, has also demonstrated that non-autoregressive TTS can be achieved without text and speech alignment. This paper can be regarded as a two-stage implementation of E2-TTS (although they have different diffusion mechanisms). With E2-TTS in mind, the findings presented in this paper are not as surprising. Additionally, this paper does not include a comparison with E2-TTS. But, E2-TTS was published at the end of June. To the best of my knowledge, papers published after July are considered concurrent work according to ICLR's review guidelines, placing E2-TTS in a grey area.

In the end, I recommend this paper for acceptance. However, I would not object if the SAC decides to reject it.

---

### Decision · Program_Chairs · 2025-01-22

Accept (Poster)